# When Models Don't Collapse:
# On the Consistency of Iterative MLE

**Daniel Barzilai**
Weizmann Institute of Science
daniel.barzilai@weizmann.ac.il

**Ohad Shamir**
Weizmann Institute of Science
ohad.shamir@weizmann.ac.il

## Abstract

The widespread use of generative models has created a feedback loop, in which each version of a model is trained on data partially produced by its predecessors. This process has raised concerns about *model collapse*: A critical degradation in performance caused by repeated training on synthetic data. However, different analyses in the literature have reached different conclusions as to the severity of model collapse. As such, it remains unclear how concerning this phenomenon is, and under which assumptions it can be avoided. To address this, we theoretically study model collapse for maximum likelihood estimation (MLE), in a natural setting where synthetic data is gradually added to the original data set. Under standard assumptions (similar to those long used for proving asymptotic consistency and normality of MLE), we establish non-asymptotic bounds showing that collapse can be avoided even as the fraction of real data vanishes. On the other hand, we prove that some assumptions (beyond MLE consistency) are indeed necessary: Without them, model collapse can occur arbitrarily quickly, even when the original data is still present in the training set. To the best of our knowledge, these are the first rigorous examples of iterative generative modeling with accumulating data that rapidly leads to model collapse.

## 1 Introduction

Generative models such as large language models (LLMs) and diffusion models are increasingly filling the internet with synthetic data. At the same time, web-scraped content remains a common source for training newer models. This creates a feedback loop in which each version of a model is trained partly on outputs of all past models, and synthetic data gradually dominates future training sets. Because identifying and filtering artificially generated content at scale is not always feasible, existing biases and artifacts in the models can become increasingly embedded in the training data and amplified from model to model. This phenomenon has recently been termed *Model Collapse*: A critical degradation in performance caused by repeated training on synthetic data [Shumailov et al., 2024, Bertrand et al., 2024, Dohmatob et al., 2024a, Gerstgrasser et al., 2025].

The effects of training on synthetic data appear to vary widely across settings. In some cases, even a small amount of synthetic data has been shown to significantly degrade performance [Dohmatob et al., 2025]; in others, model collapse can be avoided altogether [Gerstgrasser et al., 2025, Dohmatob et al., 2024a] or synthetic data may even be beneficial [Jain et al., 2024, Dohmatob et al., 2024b].

To help clarify this picture, we theoretically study the behavior of maximum likelihood estimation (MLE) under iterative training with synthetic data. We focus on a natural and practically motivated setting (also studied by Alemohammad et al. [2024], Gerstgrasser et al. [2025], Dey and Donoho [2024]), where we initially have $n$ samples from some ground-truth distribution, and at each iteration $T$, the latest model generates $n$ new samples that are then accumulated with all previous data and used to train the next model.

39th Conference on Neural Information Processing Systems (NeurIPS 2025).

Recently, Dey and Donoho [2024] analyzed a similar setting specifically for distributions arising from exponential families, and showed that for any fixed $T$ and in the limit as $n \to \infty$, the error of the iteration-$T$ model is degraded by at most a universal multiplicative constant compared to the initial estimator trained solely on real data. However, their guarantees are asymptotic in $n$ while $T$ is fixed. Therefore, they do not quantify how the performance depends on the proportion of synthetic versus real data, since the fraction of training data that is real is a constant (given by $1/T$) while $n$ grows to infinity. Therefore, it is difficult to deduce from their results how concerning model collapse may be as the fraction of real data decreases.

In contrast, in this work, we prove in Theorem 4.1 non-asymptotic guarantees that remain valid even as the fraction of real data approaches zero. Under standard regularity and smoothness assumptions (of the kind long used to prove asymptotic consistency and normality of MLE), we show that as long as the number of samples per iteration is at least *polylogarithmic* in the number of iterations, iterative MLE is consistent (meaning that it converges to the ground truth model as the sample size increases). These findings offer a sharper theoretical understanding of when model collapse can be avoided.

We complement this result with negative ones, that illustrate what can go wrong when these assumptions are violated. In particular, we construct families of distributions for which MLE is consistent when trained on real data, but nonetheless suffers from collapse when synthetic data is iteratively accumulated. Our negative results come in two flavors. In Theorem 5.1, for any fixed sample size $n$, we construct a family of distributions such that the first iteration gives an excellent approximation to the real distribution, but even the second does not with constant probability. Next, in Theorem 5.2, we show that there exists a family of distributions such that for any $n$, model collapse will eventually occur, after a number of iterations which grows arbitrarily slowly with $n$.

To the best of our knowledge, Theorem 5.1 and Theorem 5.2 are the first rigorous examples of iterative generative modeling with accumulating data that rapidly lead to model collapse. Recently, it has been suggested that model collapse does not occur when data accumulates across iterations [Gerstgrasser et al., 2025, Dey and Donoho, 2024, Schaeffer et al., 2025]. Our results show that such claims can only be true under structural assumptions beyond MLE consistency.

## 2 Related Work

Model collapse has recently drawn considerable attention, driven in part by the realization that many datasets are *already* contaminated with synthetic samples [Alemohammad et al., 2024]. A growing number of empirical studies have reported at least some level of performance degradation in models trained on such data [Shumailov et al., 2024, Alemohammad et al., 2024, Hataya et al., 2023, Bohacek and Farid, 2023, Briesch et al., 2023, Guo et al., 2023].

Several types of synthetic data contamination settings have been considered. Shumailov et al. [2024] considered a fully-synthetic setting, meaning that each model trains only on data produced by the previous model, without any real data. In such a setting, even simple Gaussian distributions can be shown to suffer from severe model collapse. In addition to a fully-synthetic setting, Alemohammad et al. [2024] considered an accumulating-data setting, where data is mixed between real and synthetic data. They observed empirically that in such cases, model collapse may either occur slowly or be avoided altogether, depending on how much real data is added at each iteration. Since then, a few works have theoretically considered accumulating-data settings [Gerstgrasser et al., 2025, Dey and Donoho, 2024]. These works suggest that data accumulation plays a significant role and can mitigate model collapse. Our results show that this is partially true: MLE in a data accumulation setting *can* avoid model collapse if the models are sufficiently well-behaved, but there *exist* models that can suffer from severe model collapse even in such a setting.

Among theoretical works, several settings have been studied that differ somewhat from the setting of this paper. A notable line of works considers linear regression, taking advantage of the closed-form expression for the least squares estimate [Dohmatob et al., 2024a, Gerstgrasser et al., 2025, Dohmatob et al., 2025]. These works focus on discriminative models, where previous models are used to label new data, not synthetically generate new data as in this paper. Moreover, the setting of [Dohmatob et al., 2025] is non-iterative, and they analyze the test error when the training data contains some samples that are labeled from a linear predictor drawn from a "bad" synthetic distribution that differs from the real one. This is a key difference from works such as Dey and Donoho [2024] that analyzed a setting where data gradually accumulates.

There are quite a few works that analyze a specific family of generative models. For example, Gaussians and kernel density estimators have been analyzed in Shumailov et al. [2024], Kazdan et al. [2024], He et al. [2025]. Fu et al. [2024] analyzed model collapse for simplified one-hidden-layer diffusion model. Dohmatob et al. [2024b] analyzed simplified token generators, including Hutter LLMs [Hutter, 2021] and associative memories [Cabannes et al., 2023]. Fu et al. [2025] analyzed several architectures under a framework they called recursive stability, which bears similarities to algorithmic stability. In contrast to all of these works, our work applies to general families of distributions.

A few works characterize iterative generative modeling by analyzing MLE as we do here. Marchi et al. [2024] assume that the differences between distributions in subsequent generations form a martingale difference sequence. However, this assumption is difficult to verify and somewhat unlikely in general. Seddik et al. [2024], Bertrand et al. [2024] analyze a setting where data is mixed from the ground truth model as well as the latest generative model. Our work focuses on a more natural setting of data accumulating over time. There is also the work of Dey and Donoho [2024], which was discussed in the introduction.

Lastly, we note that some works have proposed mechanisms to mitigate collapse through supervision or intervention. For instance, Ferbach et al. [2024], Feng et al. [2025], Amin et al. [2025] show that even minimal forms of ground-truth feedback can substantially reduce the risk of collapse. In contrast, our work focuses on the unsupervised case, where synthetic data accumulates and no corrective signal is available.

## 3 Setting and Notation

### 3.1 Notation

We use bold-faced font to denote vectors, e.g. $\mathbf{x} \in \mathbb{R}^d$, and denote by $\|\mathbf{x}\|$ the Euclidean norm. We let $[n] := \{1, \ldots, n\}$. Unless otherwise stated, $\|\cdot\|$ denotes the operator norm for matrices and 3rd-order tensors, where the latter is defined for a 3rd-order tensor $A$ as $\|A\| = \sup_{\mathbf{v}_1, \mathbf{v}_2, \mathbf{v}_3 \neq \mathbf{0}} \frac{A(\mathbf{v}_1, \mathbf{v}_2, \mathbf{v}_3)}{\|\mathbf{v}_1\| \|\mathbf{v}_2\| \|\mathbf{v}_3\|}$. We use the standard big-O notation, with $\mathcal{O}(\cdot)$, $\Theta(\cdot)$ and $\Omega(\cdot)$ hiding absolute constants that do not depend on problem parameters. To specify constants that depend only on certain quantities, we may put these quantities in parentheses. For example, $C(K_1, K_2)$ would denote a constant that depends only on $K_1, K_2$. For a given vector $\mathbf{v}$ and radius $r > 0$, we let $B_r(\mathbf{v}) := \{\mathbf{u} : \|\mathbf{u} - \mathbf{v}\| \leq r\}$ be the closed ball of radius $r$ centered at $\mathbf{v}$. For a function $f(\theta)$, we write $\nabla^2 f(\theta)$ for its Hessian, and $\nabla^3 f(\theta)$ for its 3rd-order derivative tensor, meaning $[\nabla^3 f(\theta)]_{i,j,k} = \frac{\partial^3}{\partial \theta_i \partial \theta_j \partial \theta_k} f(\theta)$. For a matrix $A$, we denote by $\lambda_{\min}(A)$, $\lambda_{\max}(A)$ its minimal and maximal eigenvalues respectively.

### 3.2 Iterative Maximum Likelihood Estimation

In this section, we formalize the iterative MLE setting that will be studied throughout the paper. Let $\Theta$ be a set of parameters and consider a corresponding family of probability density functions (PDFs) over an input space $\mathcal{X}$, given by $P_\Theta := \{p_\theta(\cdot) \mid \theta \in \Theta\}$. Generative modeling aims to approximate unknown ground truth parameters $\theta^\star$ using some $\theta \in \Theta$. Perhaps the most fundamental way to do this is through MLE (throughout the paper, we will also use this acronym to refer to the maximum likelihood *estimator* - the meaning should be clear from context).

**Definition 3.1.** *Let $X$ be a dataset with elements belonging to $\mathcal{X}$, the MLE trained on $X$ is given by*

$$\hat{\theta} := \operatorname*{argmax}_{\theta \in \Theta} \sum_{\mathbf{x} \in X} \log\left(p_\theta(\mathbf{x})\right),$$

In the above definition, it is not immediately clear why the MLE exists or if it is unique. Existence is known to hold under mild assumptions, and throughout the proofs, we will explicitly show existence whenever necessary. Regarding uniqueness, the MLE may not be unique in general. However, under mild assumptions, it is known that the MLE converges to the real parameters $\theta^\star$ (e.g. [Wald, 1949]), and that given sufficiently many samples, the log-likelihood is strictly concave in a neighborhood of $\theta^\star$. As such, asymptotically, the MLE is expected to be unique. Nevertheless, formally treating this typically introduces unnecessary and undesired complications to the analysis. It is therefore standard to simply assume that the MLE is unique whenever it exists (e.g. [Lehmann and Casella, 2006]). We

---
**Algorithm 1** Iterative Maximum Likelihood Estimation
---
**Require:** Parameter space $\Theta \subseteq \mathbb{R}^d$; family of distributions $\{p_\theta\}_{\theta \in \Theta}$ over input space $\mathcal{X}$; number of samples per iteration $n$; target parameters $\theta^\star \in \Theta$.

1: Set $\theta^{(0)} := \theta^\star$
2: **for** $T = 0, 1, 2, \ldots$ **do**
3:      sample $X^{(T)} := \{\mathbf{x}_1^{(T)}, \ldots, \mathbf{x}_n^{(T)}\} \sim p_{\theta^{(T)}}(\cdot)$ i.i.d.
4:      Define cumulative dataset: $X^{(\leq T)} := \bigcup_{t=0}^{T} X^{(t)}$
5:      Train model on $X^{(\leq T)}$:

$$\theta^{(T+1)} := \operatorname*{argmin}_{\theta \in \Theta} \sum_{t=0}^{T} \ell_t(\theta), \quad \ell_t(\theta) := -\frac{1}{n} \sum_{i=1}^{n} \log\left(p_\theta(\mathbf{x}_i^{(T)})\right),$$

6: **end for**
---

follow Bertrand et al. [2024] in making a similar, but slightly milder assumption that if there are multiple parameter vectors maximizing the log-likelihood, the argmax may choose the one that is closest to a given reference point. This is, of course, made explicit in the proofs.

Throughout the paper, we will be mostly interested in what happens when MLEs are iteratively re-trained. We will be analyzing a setting where synthetic data accumulates over time, as this is what one naturally expects to occur with web data (see the Related Works, Sec. 2, for a discussion on this). Let $\theta^\star \in \Theta$ denote the parameters of the real underlying distribution, and set $\theta^{(0)} := \theta^\star$. For each iteration $T = 0, 1, \ldots$, sample $X^{(T)} := \{\mathbf{x}_1^{(T)}, \ldots, \mathbf{x}_n^{(T)}\} \sim p_{\theta^{(T)}}(\cdot)$ i.i.d. and add these to the existing dataset, giving $X^{(\leq T)} := \bigcup_{t=0}^{T} X^{(t)}$. Then, obtain $\theta^{(T+1)}$ as the MLE given the training data $X^{(\leq T)}$. We refer the reader to Algorithm 1 for a complete description of iterative MLE. Note that for convenience, the algorithm is written as a minimization problem using the negative log likelihood (or cross-entropy loss).

We are now ready to state our assumptions. They are minor variants of those long used to study MLE in classical statistical literature (since at least Cramér [1946], see also [Le Cam, 1956, van der Vaart, 2000, Lehmann and Casella, 2006]). The first set of assumptions consists of standard regularity conditions (see e.g. [Lehmann, 1999]).

**Assumption 1** (Regularity Conditions)**.**

     *(A) There exists some $r > 0$ such that the closed ball $B_r(\theta^\star)$ is contained in $\Theta$.*

     *(B) The probability density functions $p_\theta$ are distinct.*

     *(C) The set of points for which $p_\theta$ is positive does not depend on $\theta$.*

Assumption 1. B is necessary to quantify the distance between distributions $p_\theta$, $p_{\theta'}$ using $\|\theta - \theta'\|$. Note that one can always satisfy Assumption 1. B by removing duplicates from $P_\Theta$, or by considering the quotient topology as in Redner [1981]. Assumption 1. C avoids pathologies and ensures that $\log p_\theta(x)$ is well-defined throughout the iterative sampling process. In distributions modeled using neural networks, probabilities are usually given by applying a softmax, ensuring that they are always positive and thus satisfying Assumption 1. C.

Classical analysis of MLE often require various smoothness assumptions on $\log p_\theta(\mathbf{x})$ such as bounded third derivatives (see for example [Cramér, 1946, Lehmann and Casella, 2006]). We will use the following (where $r > 0$ is the radius from Assumption 1. A):

**Assumption 2** (Smoothness)**.** *For any $\mathbf{x} \in \mathcal{X}$ and $\theta \in \Theta$, $\log(p_\theta(\mathbf{x}))$ is 3 times continuously differentiable in $\theta$, the partial derivatives support differentiation under the integral sign [1], and*

---

[1] Meaning that we can exchange the order of differentiation and integration. This is a mild assumption that is implicit in many papers.

*(A) Sub-Gaussian gradients: There exists some $K_1 > 0$ such that for any $\theta \in B_r(\theta^\star)$*

$$\mathbb{P}_{\mathbf{x}}\left(\|\nabla_\theta \log\left(p_\theta(\mathbf{x})\right)\| \geq u\right) \leq 2\exp\left(-\frac{u^2}{2K_1^2}\right), \qquad \forall u \geq 0.$$

*(B) Bounded Hessian: There exists some $K_2 > 0$ such that for any $\mathbf{x} \in \mathcal{X}$ and $\theta \in B_r(\theta^\star)$, $\left\|\nabla_\theta^2 \log\left(p_\theta(\mathbf{x})\right)\right\| \leq K_2$.*

*(C) Bounded Third Derivatives: There exists some $K_3 > 0$ such that for any $\mathbf{x} \in \mathcal{X}$ and $\theta \in B_r(\theta^\star)$, $\left\|\nabla_\theta^3 \log\left(p_\theta(\mathbf{x})\right)\right\| \leq K_3$.*

Assumptions 2. A, 2. B allow us to bound the difference between sums of random variables and their expected values. Since our bounds are non-asymptotic, one cannot avoid some assumptions to bound these differences. Sub-Gaussianity is a standard assumption in non-asymptotic works, and holds (for example) for bounded random vectors. Nevertheless, our assumptions need to hold only in a small neighborhood of $\theta^\star$, making them relatively mild. It is possible to relax these assumptions further, but we do not pursue such generalizations, as it is not the focus of our paper.

Before stating our next assumption, we recall that the Fisher information matrix at some $\theta$ is defined as

$$\mathcal{I}(\theta) := \mathbb{E}_{\mathbf{x}}\left[\nabla_\theta \log p_\theta(\mathbf{x}) \nabla_\theta \log p_\theta(\mathbf{x})^\top\right].$$

The Fisher information matrix is well-known to play a central role in the analysis of MLE. Under our other assumptions, it is straightforward to show that the Fisher information matrix is always positive semidefinite (see Appendix A for more information). In fact, standard analyses of MLE (say, to establish asymptotic normality) require the matrix to be positive definite at $\theta^\star$ [van der Vaart, 2000, Lehmann and Casella, 2006]. Thus, to get our non-asymptotic bounds, it is reasonable to assume the following (where again, $r > 0$ is the value from Assumption 1. A):

**Assumption 3.** *There exists some $\lambda_0 > 0$ such that for any $\theta \in B_r(\theta^\star)$, $\lambda_{\min}\left(\mathcal{I}(\theta)\right) \geq \lambda_0$.*

We note that one can equivalently assume that $\mathcal{I}(\theta)$ is positive definite only at $\theta^\star$, and pick $r$ small enough such that by the smoothness assumption, this holds for the neighborhood. However, the formulation above is more convenient for our purposes.

We end by noting that the assumptions above are mostly satisfied (at least approximately) by neural networks. For example, the constant support assumption (Assumption 1. C) is trivially satisfied in standard architectures, since the softmax function widely used to assign probabilities is always non-zero. The smoothness assumptions can be satisfied in various settings, especially when using techniques such as weight decay, which are standard in modern LLM training. Of course, the exact bounds would depend on the architecture and the setting. In general, we believe that weakening these assumptions is quite feasible and is an interesting direction for future work.

## 4 Consistency of Iterative MLE

In this section, we formally show that iterative MLE remains consistent under the conditions from the previous section. In particular, we provide a non-asymptotic bound, which establishes that as long as the number of samples $n$ is at least polylogarithmic in the number of iterations $T$, then with high probability, all models remain close to the ground-truth parameters. This result highlights that model collapse is not inevitable, even when $T \to \infty$ and the fraction of real data vanishes.

**Theorem 4.1.** *Under Assumptions 1 - 3, there exist constants $c, C > 0$ which depend only on $K_1, K_2, K_3, \lambda_0$ and $r$, such that for any $T \in \mathbb{N}$, $\delta > 0$ and any $n \geq c\left(\log(T) + 1\right)^2 \log^2\left(\frac{7dT}{\delta}\right)$, it holds with probability at least $1 - \delta$ that*

$$\left\|\theta^{(T)} - \theta^\star\right\| \leq C\sqrt{\frac{\log\left(\frac{4d}{\delta}\right)}{n}}. \tag{1}$$

For sufficiently large $n$, the bound in Eq. (1) is independent of $T$, and has only a logarithmic dependence on the dimension $d$. The theorem is stated for a specific $T$, but a union bound can easily provide a similar result holding simultaneously for all $t \in [T]$, at the cost of a $\log(T)$ factor.

Under the same assumptions as in Theorem 4.1, convergence of parameters also implies convergence in KL-Divergence and convergence in total variation (TV) distance. We refer the reader to Appendix A.1 for background and details. In particular, for a suitable absolute constant $C > 0$, Theorem 4.1 implies

$$D_{\mathrm{KL}}\left(p_{\theta^\star} \,\|\, p_{\theta^{(T)}}\right) \;\leq\; C \cdot \frac{\log\left(\frac{4d}{\delta}\right)}{n} \quad,\quad \mathrm{TV}\left(p_{\theta^\star}, p_{\theta^{(T)}}\right) \;\leq\; C\sqrt{\frac{\log\left(\frac{4d}{\delta}\right)}{n}} \;.$$

We now detail some ways in which Theorem 4.1 differs from past results on model (non)-collapse. In Bertrand et al. [2024], synthetic data does not accumulate across iterations, and for each iteration $t \in [T]$, most of the training data used to train $\theta^{(t)}$ is real. The maximal fraction of synthetic data was increased in the follow-up work Ferbach et al. [2024] when assuming access to the full distribution (i.e. $n = \infty$). Similarly, Dey and Donoho [2024] first fix $T$ and then analyze the limit of $n \to \infty$. They do not provide finite sample guarantees that quantify the dependence between $T$ and $n$. Seddik et al. [2024] bounded the expected value of the TV distance for distributions over finite vocabularies. When the amount of synthetic data is sufficiently large relative to the vocabulary size, their bound scales as $\mathcal{O}\left(\sqrt{k}/n\right)$ where $k$ is the total amount of synthetic data. In the data accumulation setting, $k = (T-1)n$, in which case the bound becomes $\mathcal{O}\left(\sqrt{T/n}\right)$.

We note that while Theorem 4.1 considers a setting where the exact MLE is computable, we believe the theorem can naturally be extended to also accommodate an optimization error, where only an approximate MLE is available. Indeed, we empirically observe in Appendix G that for families of distributions for which an exact formula for the MLE is known, the results are robust to mild optimization error.

## 4.1 Proof Sketch of Theorem 4.1

We provide here the proof intuition for Theorem 4.1, and refer the reader to Appendix D for the rigorous proof.

As a preliminary stage, we first show using Proposition C.1 that given enough samples, for any $t \in [T]$, $\left\|\theta^{(t+1)} - \theta^{(t)}\right\|$ is small with high probability. The challenges of this step are that this is done in a non-asymptotic way and takes into account data arising from all previous iterations.

We note that Theorem 4.1 cannot be obtained naively as a direct consequence of the Proposition C.1. Extending Proposition C.1 to a bound on $\left\|\theta^{(T)} - \theta^{(0)}\right\|$ using the triangle inequality leads to a suboptimal dependence on $T$, since it doesn't take into account cancellations from iteration to iteration. Instead, as we will show in the following paragraph, Proposition C.1 will be used to ensure that for large $n$, $\theta^{(t+1)}$ will be sufficiently close to $\theta^{(t)}$ to enable Taylor expanding the log likelihood around it. This idea draws inspiration from the asymptotic normality analysis of MLE [Cramér, 1946, Lehmann and Casella, 2006].

To that end, fix some $t \in [T]$ and observe that for any such $t$, since $\theta^{(t+1)}$ is the MLE on $X^{(\leq t)}$, it is a stationary point of the log-likelihood function. As such, Taylor expanding, we show that there exists a matrix $R_t \in \mathbb{R}^{m \times m}$ with $\|R_t\| \leq t\epsilon$ such that

$$0 = \sum_{j=0}^{t} \nabla \ell_j\left(\theta^{(t+1)}\right) = \left(\sum_{j=0}^{t} \nabla \ell_j\left(\theta^{(t)}\right) + \nabla^2 \ell_j(\theta^{(t)}) \cdot (\theta^{(t+1)} - \theta^{(t)})\right) + R_t(\theta^{(t+1)} - \theta^{(t)}).$$

By definition, $\theta^{(t)}$ is the MLE for $X^{(\leq t-1)}$, so it is a stationary point for the corresponding log-likelihood function and thus $\sum_{j=0}^{t-1} \nabla \ell_j\left(\theta^{(t)}\right) = 0$. For notational simplicity, let $H_t := \left(\sum_{j=0}^{t} \nabla^2 \ell_j(\theta^{(t)})\right) + R_t$, then the above simplifies to

$$0 = \nabla \ell_t\left(\theta^{(t)}\right) + H_t(\theta^{(t+1)} - \theta^{(t)}) \;.$$

In the full proof, we show that $H_t$ is invertible. In such a case, we can rearrange the above equation to obtain

$$\theta^{(t+1)} - \theta^{(t)} = -H_t^{-1} \nabla \ell_t \left( \theta^{(t)} \right).$$

Importantly, this allows us to express how the parameters evolve over many iterations by taking a telescopic sum as follows.

$$\left\| \theta^{(T)} - \theta^{(0)} \right\| = \left\| \sum_{t=0}^{T-1} \theta^{(t+1)} - \theta^{(t)} \right\| = \left\| \sum_{t=0}^{T-1} H_t^{-1} \nabla \ell_t \left( \theta^{(t)} \right) \right\|$$

$$\leq \frac{1}{\lambda_0} \left\| \sum_{t=0}^{T-1} \frac{1}{t+1} \nabla \ell_t \left( \theta^{(t)} \right) \right\| + \left\| \sum_{t=0}^{T-1} \left( H_t^{-1} - \frac{1}{t+1} \mathcal{I}(\theta^{(0)})^{-1} \right) \nabla \ell_t \left( \theta^{(t)} \right) \right\|. \tag{2}$$

The expected value of $\nabla \ell_t \left( \theta^{(t)} \right)$ (conditioned on $\theta^{(t)}$) can be shown to be zero, so that the first term forms a martingale, which allows us to bound the norm essentially as if all samples were independent. Since each $\nabla \ell_t$ is scaled by $\frac{1}{t+1}$, the variance scales as $\frac{1}{(t+1)^2}$. So the variance of the sum can be upper bounded as $\sum_{t=1}^{T} \frac{1}{t^2} \leq \sum_{t=1}^{\infty} \frac{1}{t^2} \leq \frac{\pi^2}{6}$. In summary, we show that with high probability

$$\left\| \sum_{t=0}^{T-1} \frac{1}{t+1} \nabla \ell_t \left( \theta^{(t)} \right) \right\| \leq \mathcal{O} \left( \frac{\log \left( \frac{d}{\delta} \right)}{\sqrt{n}} \right).$$

The second term in Eq. (2) has to be treated differently, as correlations between $H_{t-1}$ and $\nabla \ell_t$ imply that each term is not necessarily mean-zero, and so the sum should be expected to have some dependence on $T$. This term somewhat complicates the proof, as bounding it requires knowing that $\left\| \theta^{(t)} - \theta^{(0)} \right\|$ is sufficiently small for all $t < T$. The proof thus works inductively, bounding this term from iteration to iteration. Roughly speaking, in the end, we show that with high probability

$$\sum_{t=0}^{T-1} \left\| H_t^{-1} - \frac{1}{t+1} \mathcal{I}(\theta^{(0)})^{-1} \right\| \cdot \left\| \nabla \ell_t \left( \theta^{(t)} \right) \right\| \leq \frac{\sqrt{c} \log(T+1) \log \left( \frac{dT}{\delta} \right)}{n} \leq \frac{1}{\sqrt{n}},$$

where the last inequality follows from the assumption that $n$ is sufficiently large.

## 5 Necessity of Structural Assumptions

Theorem 4.1 provides conditions under which the iterative MLE retains good performance, even if the proportion of synthetic data approaches 1. Clearly, this cannot always be true. In particular, there are well-known examples of families of distributions on which even standard MLE is inconsistent: Namely it will not converge to the ground-truth parameters as the sample size increases, even when trained purely on real data (e.g. [Bahadur, 1958, Ferguson, 1982, Le Cam, 1990]). In such situations, the whole question of model collapse is rather meaningless. Thus, a natural (informal) follow-up question is the following: *In the setting where synthetic data is added to the real dataset in each iteration, is there a family of distributions that is sufficiently well-behaved for MLE to be asymptotically consistent (when trained on real data), but still exhibits rapid model collapse?* In other words, do there exist cases where the MLE *can* learn the real distribution, *and yet* model collapse still occurs when applying MLE iteratively?

In this section, we show that the answer is yes, and demonstrate different settings in which model collapse can occur when the conditions of Theorem 4.1 are not satisfied. To the best of our knowledge, these are the first rigorous examples of iterative generative modeling with accumulating data that rapidly leads to model collapse.

We emphasize that, following the rest of the paper, we focus here on a setting where synthetic data iteratively accumulates on top of the real data. A different model collapse setting studied in some previous works is when at each iteration, MLE is performed purely on synthetic data generated by the latest model. In such a setting, the real training data disappears already after a single iteration, and it

has been shown to lead to model collapse even for very well-behaved distributions such as Gaussians [Shumailov et al., 2024]. It has recently been suggested that if data is added rather than replaced (as in our setting), the extent to which iterative MLE performance degrades is limited [Gerstgrasser et al., 2025, Dey and Donoho, 2024, Schaeffer et al., 2025]. We show here that this can be true only if further assumptions are made, beyond just MLE consistency (as we do in Theorem 4.1).

To formalize our results, we will require the following consistency definition for MLE:

**Definition 5.1.** *We will say a family of distributions $P_\Theta$ is TV-consistent, if for any $\theta^\star \in \Theta$ and $n \in \mathbb{N}$, the MLE $\hat{\theta}$ trained on $n$ i.i.d. samples from $p_{\theta^\star}$ exists, and*

$$\mathrm{TV}\left(p_{\theta^\star}, p_{\hat{\theta}}\right) \xrightarrow[n \to \infty]{\mathbb{P}} 0.$$

Note that we use here convergence in total variation, rather than convergence in parameters as in Theorem 4.1. The reason is that to establish our negative results, we have to make use of distributions that do not follow the assumptions of Theorem 4.1, and in particular do not satisfy the smoothness assumptions there. Without smoothness, parametric convergence and convergence of distributions are no longer equivalent in general. Thus, using a probability metric such as total variation is more natural in our setting, as we are ultimately interested in approximating the ground-truth distribution.

## 5.1 Models Can Collapse Immediately

By definition, for a TV-consistent family of distributions, $p_{\theta^{(1)}}$ is a good approximation of the ground truth distribution $p_{\theta^\star}$, assuming the number of samples $n$ is sufficiently large. Our first negative result shows that, perhaps surprisingly, one cannot hope to show the same even for $p_{\theta^{(2)}}$ without further assumptions. Specifically, for any $n$ there is some family of distributions (that may depend on $n$), such that MLE on $n$ samples from the ground-truth distribution will perform well, but if we now augment the data with $n$ synthetic samples from the MLE solution, and re-run MLE, then the resulting distribution $p_{\theta^{(2)}}$ will exhibit model collapse with constant probability.

**Theorem 5.1.** *There exists $\Theta \subseteq \mathbb{R}^2$ and $\theta^\star \in \Theta$, such that for any $n \in \mathbb{N}$, there is a TV-consistent family of distributions $\{p_\theta\}_{\theta \in \Theta}$ (that may depend on $n$) such that*

  *1. with probability at least $1 - \frac{1}{n}$,*

$$\mathrm{TV}\left(p_{\theta^\star}, p_{\theta^{(1)}}\right) \leq \frac{\log(n)}{n}.$$

  *2. For some absolute constants $c, C > 0$, it holds with probability at least $c$ that*

$$\mathrm{TV}\left(p_{\theta^\star}, p_{\theta^{(2)}}\right) \geq C.$$

In the above theorem, as the number of samples grows, we can find a family of distributions such that $p_{\theta^{(1)}}$ is very close to $p_{\theta^\star}$ with high probability, but there is some constant probability that $p_{\theta^{(2)}}$ will be far from $p_{\theta^{(1)}}$. This implies that statements similar to Theorem 4.1 are not possible for general TV-consistent families without further assumptions. Indeed, Theorem 5.1 implies that the relative gap in total variation between iterations $t = 1$ and $t = 2$ can be arbitrarily large, since

$$\frac{\mathrm{TV}\left(p_{\theta^\star}, p_{\theta^{(2)}}\right)}{\mathrm{TV}\left(p_{\theta^\star}, p_{\theta^{(1)}}\right)} \geq C \cdot n / \log(n).$$

We now provide some intuition for the proof of Theorem 5.1, with the full rigorous proof appearing in Appendix E. We consider a family of distributions, given by the following parameterized mixture of uniform distributions on $\mathbb{R}$:

$$\frac{1}{2} \cdot U([0,1]) + \frac{1-\alpha}{2} \cdot U([0, 1-2\alpha]) + \frac{\alpha}{4} \cdot U([2,3]) + \frac{\alpha}{4} \cdot U([\mu, \mu + f(\alpha)]),$$

where $U(\cdot)$ is the uniform distribution on an interval, $\Theta = \left\{(\alpha, \mu) \mid \alpha \in \left[0, \frac{1}{4}\right] \mu \in [2, 3 - f(\alpha)]\right\}$ are the parameters, and $f$ is a positive function that decays very quickly with $\alpha$, so that the PDF of $U([\mu, \mu + f(\alpha)])$ approaches a delta function (the exact form of $f$ depends on $n$). Let $\theta^{(0)} := \theta^\star := (\alpha^{(0)} = 0, \mu^{(0)} = 0)$ such that $p_{\theta^{(0)}} = U([0,1])$. We show that the MLE $\theta^{(1)} = (\alpha^{(1)}, \mu^{(1)})$ satisfies

$$\alpha^{(1)} = \frac{1 - \max_{i \in [n]} x_i^{(0)}}{2} \approx \frac{1}{2n}.$$

As such, $\alpha^{(1)}$ converges very quickly to $\alpha^{(0)}$ as $n$ increases, and we prove that this implies a rapid convergence of TV $(p_{\theta^{(0)}}, p_{\theta^{(1)}})$, regardless of the value of $\mu^{(1)}$.

We now move on to analyzing the second iteration. Because $\alpha^{(1)} \approx \frac{1}{2n}$, then with some constant probability (over the sampling of $n$ new samples from $p_{\theta^{(1)}}$), at least one of these samples $x_i^{(1)}$ will be inside the interval $[2, 3]$. When this happens, because $f(\alpha)$ is tiny for larger values of $\alpha$ (leading to a high likelihood in the interval $[\mu, \mu + f(\alpha)]$), the MLE solution $\theta^{(2)} = (\alpha^{(2)}, \mu^{(2)})$ will be such that $x_i^{(1)} \in [\mu^{(2)}, \mu^{(2)} + f(\alpha^{(2)})]$ and $\alpha^{(2)}$ will be sufficiently large so that $f(\alpha^{(2)})$ is very small. In particular, $\alpha^{(2)}$ will be considerably larger than the ground truth $\alpha^{(0)} = 0$, leading to model collapse.

## 5.2 Arbitrarily Fast Model Collapse

Theorem 5.1 shows that without further assumptions, model collapse can occur already after a single iteration. However, the construction requires picking the distribution according to the sample size $n \in \mathbb{N}$, which is arguably unnatural. Below, we show that this requirement can be removed: Namely, there exists a family of distributions where model collapse will occur for any sample size $n$. On the flip side, the model collapse no longer occurs after a single iteration, but rather after a number of iterations that grows with $n$ (although the growth rate can be arbitrarily slow):

**Theorem 5.2.** *Let $\phi : (0, \infty) \to (0, \infty)$ be any strictly monotonically increasing function such that $\lim_{n \to \infty} \phi(n) = \infty$. Then there exists an absolute constant $C > 0$, a set $\Theta$, $\theta^\star \in \Theta$ and a TV-consistent family of distributions $P_\Theta$ (which depends on $\phi$), such that for any $\delta \in (0, 1)$, $n \in \mathbb{N}$, it holds with probability at least $1 - \delta$ that*

$$\text{TV} \left( p_{\theta^{(T)}}, p_{\theta^\star} \right) \geq \frac{3}{8} \quad \text{for some} \quad T \leq \left\lceil \frac{C}{\delta} \log \left( \frac{4}{\delta} \right) \cdot \max \left( \phi(n), 1 \right) \right\rceil .$$

Importantly, $\phi$ can be chosen to grow arbitrarily slowly. For example, taking $\phi(n) = \log \log(n + 1)$, Theorem 5.2 implies that one can exhibit model collapse in as few as $\mathcal{O}(\log \log(n + 1))$ iterations.

The proof of Theorem 5.2 draws inspiration from the proof of Theorem 5.1, but the construction is more involved, as the distribution can no longer depend on the number of samples $n$. The family will consist of two types of distributions. The first, which we denote as $h_{\boldsymbol{\alpha}}$, has the form

$$\sum_{j=0}^{\infty} (1 - \alpha_j) \left( \prod_{k=0}^{j-1} \alpha_k \right) U([j, j + 1 - 2\alpha_j]) ,$$

where $\alpha_i \in [0, \frac{1}{4}]$, and $\boldsymbol{\alpha}$ has a finite number of non-zero indices. One way to think of these distributions is as sampling using an iterative process, where starting from $j = 0$, one flips a coin with bias $\alpha_j$, and either samples a point from $U([j, j + 1 - 2\alpha_j])$ (with probability $1 - \alpha_j$), or with probability $\alpha_j$, increase $j$ by one and repeat the process, until some point is sampled. We also include a family of distributions $g_{\beta, J}$ corresponding to

$$\frac{1}{2} U[0, J] + \frac{1}{2} U([J - \beta, J - \beta + f(J)]) ,$$

where $J \in \mathbb{N} \setminus \{1\}$, $\beta \in [0, 1]$ and $f$ is a function that decays very quickly as $J$ increases.

Now, consider the ground truth distribution to be $h_{\mathbf{0}}$ (meaning $\alpha_j^{(0)} = 0$ for all $j$), which is actually just $U([0, 1])$. We show that at any iteration $t$, the density $h_{\boldsymbol{\alpha}^{(t)}}$ that maximizes the likelihood out of functions of the form $h_{\boldsymbol{\alpha}}$ is given by taking $\alpha_j^{(t+1)} = \frac{1}{2}(1 - \max X^{(\leq t)} \cap [j, j + 1] - j)$.

Thus, the general procedure is as follows: For any $J \in \mathbb{N}$, once $\alpha_j^{(t)} > 0$ for every $j \leq J - 1$, there is a non-zero chance that a new sample $x_i^{(t)}$ will reach interval $[J, J + 1]$, ensuring $\alpha_J^{(t+1)} > 0$. We choose the function $f$ so that for any $N \in \mathbb{N}$, there is some $J_N$ such that if $n \leq N$ and if there is some sample in $[J_N, J_N + 1]$, then the MLE will be of the form $g_{\beta, J}$ (as $f(J_N)$ is sufficiently small, leading to a high likelihood of the sample). We show that once this happens, the total variation distance will be large, and the proof will be complete.

The difficult part is showing that for any $J \in \mathbb{N}$, there is some time $T \in \mathbb{N}$ such that with high probability, there will be a sample in $[J, J + 1]$. Moreover, this $T$ can be chosen to be essentially

independent of $n$. Since we may let $J_N$ grow arbitrarily slowly in $N$, and the number of iterations needed to obtain a sample in $[J_N, J_N + 1]$ can be upper bounded independently of $N$, the number of iterations needed for model collapse can grow arbitrarily slowly with $N$ (and thus with $n$).

### 5.3 Implications and Relation to Theorem 4.1

The results of this section inform us how in the absence of the assumptions of Theorem 4.1, model collapse can occur arbitrarily quickly. Even though the constructions of Theorems 5.1 and 5.2 are artificial, they highlight more general phenomena that are needed for model collapse to occur or to be avoided. In particular, in our view, the main difference from Theorem 4.1 is the smoothness assumption. Theorems 5.1 and 5.2 crucially use a highly non-smooth construction, in which slight perturbations of the parameters can induce huge differences in the resulting model, and we find it unlikely that a negative example would be possible without this behavior.

## 6   Discussion

We studied model collapse in a setting that has recently gained interest in the literature, where synthetic data accumulates over time. Focusing on MLE, we showed that collapse can be avoided under standard assumptions even as the proportion of real data vanishes, provided that the number of samples is polylogarithmic in the number of iterations. At the same time, when these assumptions are not satisfied, we construct scenarios where the MLE is consistent, yet collapse occurs arbitrarily quickly with synthetic data. These examples show that MLE consistency alone is not sufficient for preventing model collapse even in the accumulating-data setting.

While the assumptions in this work are rather classic, they may not be the mildest possible while still allowing for positive results. Moving forward, it would be interesting to bridge the gap still present in this work between the assumptions in the negative and positive results and characterize assumptions that are both necessary and sufficient for avoiding model collapse. Our hope is that these results contribute to a clearer theoretical understanding of model collapse, and lead to a more fine-grained perspective on when it does or does not occur.

## Acknowledgments and Disclosure of Funding

This research is supported in part by European Research Council (ERC) grant 754705, by the Israeli Council for Higher Education (CHE) via the Weizmann Data Science Research Center and by research grants from the Estate of Harry Schutzman and the Anita James Rosen Foundation.

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

# A  Background on Likelihood Estimation

For any $\theta \in \Theta$, the Fisher information matrix is defined as

$$\mathcal{I}(\theta) := \mathbb{E}_{\mathbf{x}} \left[ \nabla_\theta \log p_\theta(\mathbf{x}) \nabla_\theta \log p_\theta(\mathbf{x})^\top \right].$$

We state here some well-known results regarding the Fisher information matrix that will be used throughout the proofs (e.g. [Lehmann, 1999][Section 7.5]).

**Theorem A.1.** *If Assumptions 1, 2 hold, then for any $\theta \in B_r(\theta^{(0)})$,*

$$\mathbb{E}_{\mathbf{x}} \left[ \nabla_\theta \log p_\theta(\mathbf{x}) \right] = 0. \tag{3}$$

Note that in particular, this implies that $\mathcal{I}(\theta)$ is the covariance matrix of the random vector $\nabla_\theta \log p_\theta(\mathbf{x})$ and is therefore p.s.d.

We will also need the following.

**Theorem A.2.** *If Assumptions 1, 2 hold, then for any $\theta \in B_r(\theta^{(0)})$,*

$$\mathcal{I}(\theta) = -\mathbb{E}_{\mathbf{x}} \left[ \nabla_\theta^2 \log p_\theta(\mathbf{x}) \right]. \tag{4}$$

Note that the above theorem also implies $\|\mathcal{I}(\theta)\| \leq \sup_{\mathbf{x}} \left\| \nabla_\theta^2 \log p_\theta(\mathbf{x}) \right\| \leq K_2$. We state this formally as the following corollary.

**Corollary A.1.** *If Assumptions 1, 2 hold, then for any $\theta \in B_r(\theta^{(0)})$,*

$$\|\mathcal{I}(\theta)\| \leq K_2. \tag{5}$$

## A.1  Parametric Convergence vs. KL vs. TV

Two common ways to compare PDFs $p, q$ over an input space $\mathcal{X}$ are the KL divergence:

$$D_{\mathrm{KL}} \left( p \,\|\, q \right) := \mathbb{E}_{x \sim p} \left[ \log \left( \frac{p(x)}{q(x)} \right) \right],$$

and the total variation distance

$$\mathrm{TV} \left( p, q \right) := \frac{1}{2} \int_{x \in \mathcal{X}} |p(x) - q(x)| \, dx.$$

We note that while the TV is a proper metric, the KL divergence is not, as it is not symmetric. Nevertheless, the two can be related by the well-known Pinsker's inequality:

$$\mathrm{TV} \left( p, q \right) \leq \sqrt{\frac{1}{2} D_{\mathrm{KL}} \left( p \,\|\, q \right)}.$$

It is well known that under sufficient smoothness assumptions, convergence in parameters implies convergence in KL and total variation. Indeed, for a fixed $\mathbf{x} \in \mathcal{X}$, consider a second-order Taylor expansion of $\log \left( p_\theta(\mathbf{x}) \right)$ around $\theta^{(0)}$, which gives

$$\log \left( p_{\theta^{(0)}}(\mathbf{x}) \right) + \nabla \log \left( p_{\theta^{(0)}}(\mathbf{x}) \right)^\top \left( \theta - \theta^{(0)} \right) + \frac{1}{2} \left( \theta - \theta^{(0)} \right)^\top \nabla^2 \log \left( p_{\theta^{(0)}}(\mathbf{x}) \right) \left( \theta - \theta^{(0)} \right) + R(\mathbf{x}),$$

where the remainder $R(\mathbf{x})$ can be shown to satisfy $|R(\mathbf{x})| \leq \frac{K_3}{6} \left\| \theta - \theta^{(0)} \right\|^3$ under Assumption 2. By Theorem A.1 the expected value of the gradient term is 0 and by Theorem A.2 the expected value of the hessian term is $-\mathcal{I}(\theta^{(0)})$.

As such, Taylor expanding at every point $\mathbf{x}$ together with Theorem A.1 and Theorem A.2, the KL divergence can be approximated as

$$D_{\mathrm{KL}}\left(p_{\theta^{(0)}} \,\|\, p_\theta\right) = \mathbb{E}_{\mathbf{x} \sim p_{\theta^{(0)}}}\left[\log\left(p_{\theta^{(0)}}(\mathbf{x})\right) - \log\left(p_\theta(\mathbf{x})\right)\right]$$

$$= \frac{1}{2}(\theta - \theta^{(0)})^\top \mathcal{I}(\theta^{(0)})(\theta - \theta^{(0)}) - \mathbb{E}[R(\mathbf{x})]$$

$$\leq \frac{1}{2}\left\|\mathcal{I}(\theta^{(0)})\right\| \cdot \left\|\theta - \theta^{(0)}\right\|^2 + \frac{K_3}{6}\left\|\theta - \theta^{(0)}\right\|^3$$

$$\leq \frac{K_2}{2}\left\|\theta - \theta^{(0)}\right\|^2 + \frac{K_3}{6}\left\|\theta - \theta^{(0)}\right\|^3.$$

By Pinsker's inequality, this implies

$$\mathrm{TV}\left(p_{\theta^{(0)}}, p_\theta\right) \leq \sqrt{\frac{1}{2}D_{\mathrm{KL}}\left(p_{\theta^{(0)}} \,\|\, p_\theta\right)} \leq \sqrt{\frac{K_2}{4}\left\|\theta - \theta^{(0)}\right\|^2 + \frac{K_3}{12}\left\|\theta - \theta^{(0)}\right\|^3}$$

$$\leq \frac{\sqrt{K_2}}{2}\left\|\theta - \theta^{(0)}\right\| + \sqrt{\frac{K_3}{12}}\left\|\theta - \theta^{(0)}\right\|^{\frac{3}{2}}.$$

# B    Concentration

We start with a couple of known results that will be useful for approximating the gradient and hessian of the log-likelihood.

**Theorem B.1** (Jin et al. [2019] Corollary 7). *Let $\mathbf{z}_1, \ldots, \mathbf{z}_T \in \mathbb{R}^d$ be random vectors and assume there exist fixed $\sigma_1, \ldots, \sigma_t$ such that for all $t \in [T]$, $\mathbb{E}[\mathbf{z}_t \mid \mathbf{z}_1, \ldots, \mathbf{z}_{t-1}] = \mathbf{0}$ and*

$$\mathbb{P}\left(\|\mathbf{z}_t\| \geq u \mid \mathbf{z}_1, \ldots, \mathbf{z}_{t-1}\right) \leq 2\exp\left(-\frac{u^2}{2\sigma_t^2}\right), \qquad \forall u \geq 0.$$

*Then there exists an absolute constant $C > 0$ such that for any $\delta > 0$, with probability at least $1 - \delta$,*

$$\left\|\sum_{t=1}^T \mathbf{z}_t\right\| \leq C\sqrt{\sum_{t=1}^T \sigma_t^2 \log\left(\frac{2d}{\delta}\right)}.$$

**Theorem B.2** (Tropp [2012] Theorem 7.1). *Let $\{M_t\}$ be a finite sequence of random symmetric $d \times d$ matrices such that $\mathbb{E}[M_t \mid M_1, \ldots, M_{t-1}] = \mathbf{0}$. Assume further that there exists a fixed sequence of symmetric $d \times d$ matrices $\{A_t\}$ such that $M_t^2 \preceq A_t^2$ almost surely. Let $\sigma^2 := \left\|\sum_t A_t^2\right\|$, then for all $u \geq 0$,*

$$\mathbb{P}\left(\lambda_{\max}\left(\sum_t M_t\right) \geq u\right) \leq d\exp\left(-\frac{u^2}{8\sigma^2}\right).$$

We bring Theorem B.3 to a slightly more convenient form for our uses.

**Theorem B.3.** *Let $\{M_t\}_{t=1}^T$ be a finite sequence of random symmetric $d \times d$ matrices such that $\mathbb{E}[M_t \mid M_1, \ldots, M_{t-1}] = \mathbf{0}$. Assume further that there exists some $K > 0$ such that $\|M_t\| \leq K$ almost surely. Then for any $\delta > 0$, it holds with probability at least $1 - \delta$ that*

$$\left\|\sum_{t=1}^T M_t\right\| \leq \sqrt{8}K\sqrt{T\log\left(\frac{2d}{\delta}\right)}.$$

*Proof.* Set $A_t^2 := K^2 I_d$, $\sigma^2 := TK^2$, apply Theorem B.2 once to bound $\sum_t M_t$ and again to bound $-\sum_t M_t$. The corollary follows from the union bound. □

**Lemma B.1.** *Under Assumptions 1, 2, if $\theta^{(1)}, \ldots, \theta^{(T-1)} \in B_r(\theta^{(0)})$ then there exists an absolute constant $C > 0$ such that for any $\delta > 0$, it holds with probability at least $1 - \delta$ that*

$$\left\|\sum_{t=0}^{T-1} \frac{1}{t+1}\nabla \ell_t(\theta^{(t)})\right\| \leq CK_1\sqrt{\frac{\log\left(\frac{2d}{\delta}\right)}{n}}.$$

*Proof.* For $t \in \{0, \ldots, T-1\}$, $i \in [n]$ let $\mathbf{z}_{t,i} := \frac{1}{t+1} \nabla \log \left( p_{\theta^{(t)}}(\mathbf{x}_i^{(t)}) \right)$. We order these $Tn$ random vectors $\mathbf{z}_{t,i}$ first by $t$ and then by $i$. Specifically, let $\rho : [Tn] \to \{0, \ldots, T-1\} \times [n]$ be this mapping of indices, such that

$$\mathbf{z}_{\rho(1)}, \ldots, \mathbf{z}_{\rho(Tn)} := \mathbf{z}_{0,1}, \ldots, \mathbf{z}_{0,n}, \mathbf{z}_{1,1}, \ldots, \mathbf{z}_{1,n}, \ldots, \mathbf{z}_{T-1,n}.$$

By Theorem A.1, for all $k \in [Tn]$, $\mathbb{E}[\mathbf{z}_{\rho(k)} \mid \mathbf{z}_{\rho(1)}, \ldots, \mathbf{z}_{\rho(k-1)}] = \mathbf{0}$. Furthermore, by Assumption 2. A, for any $t, i$ and any $u \geq 0$

$$\mathbb{P} \left( \left\| \frac{1}{t+1} \nabla \log \left( p_{\theta^{(t)}}(\mathbf{x}_i^{(t)}) \right) \right\| \geq u \right) = \mathbb{P} \left( \left\| \nabla \log \left( p_{\theta^{(t)}}(\mathbf{x}_i^{(t)}) \right) \right\| \geq (t+1)u \right)$$
$$\leq 2 \exp \left( -\frac{(t+1)^2 u^2}{2K_1^2} \right).$$

In particular, for all $k \in [Tn]$ letting $\sigma_k := K_1/(\rho(k)_1 + 1)$ (where $\rho(k)_1$ is the $t \in \{0, \ldots, T-1\}$ that corresponds to $\rho(k)$) we have

$$\mathbb{P} \left( \|\mathbf{z}_k\| \geq u \mid \mathbf{z}_1, \ldots, \mathbf{z}_{k-1} \right) \leq 2 \exp \left( -\frac{u^2}{2\sigma_k^2} \right), \qquad \forall u \geq 0.$$

As such, by Theorem B.1 there exists an absolute constant $C > 0$ such that with probability at least $1 - \delta$,

$$\left\| \sum_{k=1}^{Tn} \mathbf{z}_{\rho(k)} \right\| \leq C \sqrt{\sum_{k=1}^{Tn} \sigma_t^2 \log \left( \frac{2d}{\delta} \right)}.$$

Note that since $\sum_{t=1}^{T} \frac{1}{t^2} \leq \frac{\pi^2}{6}$, we have

$$\sum_{k=1}^{Tn} \sigma_k^2 = K_1^2 \sum_{i=1}^{n} \sum_{t=1}^{T} \frac{1}{t^2} \leq \frac{\pi^2}{6} K_1^2 n.$$

We obtain with the same probability that for a suitable altered constant $C > 0$,

$$\left\| \sum_{t=0}^{T-1} \frac{1}{t+1} \nabla \ell_t(\theta^{(t)}) \right\| = \frac{1}{n} \left\| \sum_{k=1}^{Tn} \mathbf{z}_{\rho(k)} \right\| \leq CK_1 \sqrt{\frac{\log \left( \frac{2d}{\delta} \right)}{n}}.$$

$\square$

We can also obtain concentration for a single $\bar{\theta} \in B_r(\theta^{(0)})$. We omit the proof as it is a simplified version of Lemma B.1 (specifically, the assumptions and Theorem A.1 imply that the conditions of Theorem B.1 are satisfied, which gives the following result).

**Lemma B.2.** *Let $\bar{\theta} \in B_r(\theta^{(0)})$ and $\mathbf{x}_1, \ldots, \mathbf{x}_n \sim p_{\bar{\theta}}$ i.i.d. Under Assumptions 1, 2, there exists an absolute constant $C > 0$ such that for any $\theta \in B_r(\theta^{(0)})$, $\delta > 0$, it holds with probability at least $1 - \delta$ that*

$$\left\| \frac{1}{n} \sum_{i=1}^{n} \nabla \log \left( p_\theta(\mathbf{x}_i) \right) \right\| \leq CK_1 \sqrt{\frac{\log \left( \frac{2d}{\delta} \right)}{n}}.$$

We will also need the following result for the Hessian of the log-likelihood:

**Lemma B.3.** *Under Assumptions 1, 2, if $\theta^{(1)}, \ldots, \theta^{(T-1)} \in B_r(\theta^{(0)})$ then there exists an absolute constant $C > 0$ such that for any $\delta > 0$, it holds with probability at least $1 - \delta$ that*

$$\left\| \sum_{t=0}^{T-1} \nabla^2 \ell_t(\theta^{(t)}) - \mathcal{I} \left( \theta^{(t)} \right) \right\| \leq CK_2 \sqrt{\frac{T \log \left( \frac{2d}{\delta} \right)}{n}}.$$

*Proof.* For $t \in \{0, \ldots, T-1\}$, $i \in [n]$ let $M_{t,i} := -\nabla^2 \log \left( p_{\theta^{(t)}}(\mathbf{x}_i^{(t)}) \right) - \mathcal{I}(\theta^{(t)})$. We order these $Tn$ random matrices $M_{t,i}$ first by $t$ and then by $i$. Specifically, let $\rho : [Tn] \to \{0, \ldots, T-1\} \times [n]$ be this mapping of indices, such that

$$M_{\rho(1)}, \ldots, M_{\rho(Tn)} := M_{0,1}, \ldots, M_{0,n}, M_{1,1}, \ldots, M_{1,n}, \ldots, M_{T-1,n}.$$

By Theorem A.2, for all $k \in [Tn]$, $\mathbb{E}[M_{\rho(k)} \mid M_{\rho(1)}, \ldots, M_{\rho(k-1)}] = \mathbf{0}$. Furthermore, by Assumption 2. B and Corollary A.1, for any $k \in [Tn]$,

$$\left\| M_{\rho(k)} \right\| \le K_2 + \left\| \mathcal{I}(\theta^{(t)}) \right\| \le 2K_2.$$

As such, by Theorem B.3 there exists an absolute constant $C > 0$ such that with probability at least $1 - \delta$,

$$\left\| \sum_{t=0}^{T-1} \nabla^2 \ell_t(\theta^{(t)}) - \mathcal{I}\left(\theta^{(t)}\right) \right\| = \frac{1}{n} \left\| \sum_{k=1}^{Tn} M_{\rho(k)} \right\| \le CK_2 \sqrt{\frac{T \log\left(\frac{2d}{\delta}\right)}{n}}.$$

$\square$

Once again, we can also obtain an analogous result for a single $\bar{\theta} \in B_r(\theta^{(0)})$. The proof is also analogous to Lemma B.3.

**Lemma B.4.** *Let $\bar{\theta} \in B_r(\theta^{(0)})$ and $\mathbf{x}_1, \ldots, \mathbf{x}_n \sim p_{\bar{\theta}}$ i.i.d. Under Assumptions 1, 2, there exists an absolute constant $C > 0$ such that for any $\theta \in B_r(\theta^{(0)})$, $\delta > 0$, it holds with probability at least $1 - \delta$ that*

$$\left\| -\frac{1}{n} \sum_{i=1}^{n} \nabla^2 \log \left( p_{\bar{\theta}}(\mathbf{x}_i) \right) - \mathcal{I}\left(\bar{\theta}\right) \right\| \le CK_2 \sqrt{\frac{\log\left(\frac{2d}{\delta}\right)}{n}}.$$

## C   Preparatory Results

### C.1   Non-Asymptotic Consistency

**Lemma C.1.** *If Assumption 2 holds, then for every $x \in \mathcal{X}$, $\nabla_\theta^2 \log p_\theta(\mathbf{x})$ is $K_3$-Lipschitz on $B_r(\theta^{(0)})$; that is,*

$$\left\| \nabla_\theta^2 \log p_\theta(\mathbf{x}) - \nabla_\theta^2 \log p_{\theta'}(\mathbf{x}) \right\| \le K_3 \left\| \theta - \theta' \right\|, \quad \forall \theta, \theta' \in B_r(\theta^{(0)}).$$

*Proof.* Fix $\mathbf{x} \in \mathcal{X}$ and $\theta, \theta' \in B_r(\theta^{(0)})$. Consider the line segment $\gamma : [0,1] \to B_r(\theta^{(0)})$ given by $\gamma(t) = \theta + t(\theta' - \theta)$. Note that the convexity of $B_r(\theta^{(0)})$ implies that $\gamma(t) \in B_r(\theta^{(0)})$ for all $t \in [0,1]$. From the fundamental theorem of calculus,

$$\nabla_\theta^2 \log p_{\theta'}(\mathbf{x}) - \nabla_\theta^2 \log p_\theta(\mathbf{x}) = \int_0^1 \frac{d}{dt} \nabla_\theta^2 \log p_{\gamma(t)}(\mathbf{x}) dt = \int_0^1 \nabla_\theta^3 \log p_{\gamma(t)}(\mathbf{x})[\theta' - \theta] dt,$$

where $\left[ \nabla_\theta^3 \log p_{\gamma(t)}(\mathbf{x})[\theta' - \theta] \right]_{ij} = \sum_{k=1}^d \frac{\partial^3}{\partial \theta_i \partial \theta_j \partial \theta_k} \log p_{\gamma(t)}(\mathbf{x})[\theta' - \theta]_k$.

Applying the operator norm and Assumption 2,

$$\begin{aligned} \left\| \nabla_\theta^2 \log p_{\theta'}(\mathbf{x}) - \nabla_\theta^2 \log p_\theta(\mathbf{x}) \right\| &\le \int_0^1 \left\| \nabla_\theta^3 \log p_{\gamma(t)}(\mathbf{x})[\theta' - \theta] \right\| dt \\ &\le \sup_{t \in [0,1]} \left\| \nabla_\theta^3 \log p_{\gamma(t)}(\mathbf{x}) \right\| \cdot \left\| \theta' - \theta \right\| \\ &\le K_3 \left\| \theta' - \theta \right\|. \end{aligned}$$

$\square$

**Lemma C.2.** *Under Assumptions 1, 2, for any $t \in \mathbb{N}$, if $\theta^{(0)}, \ldots, \theta^{(t)} \in B_r(\theta^{(0)})$, then there exists an absolute constant $C > 0$ such that for any $\delta > 0$, with probability at least $1 - \delta$*

$$\left\| \sum_{j=0}^{t} \nabla^2 \ell_j(\theta^{(t)}) - \mathcal{I}(\theta^{(j)}) \right\| \leq C(K_2 + K_3) \left( \sqrt{\frac{(t+1) \log\left(\frac{2d}{\delta}\right)}{n}} + t \max_{j \leq t} \left\| \theta^{(j)} - \theta^{(0)} \right\| \right).$$

*Proof.* By the triangle inequality,

$$\left\| \sum_{j=0}^{t} \nabla^2 \ell_j(\theta^{(t)}) - \mathcal{I}(\theta^{(j)}) \right\| \leq \left\| \sum_{j=0}^{t} \nabla^2 \ell_j(\theta^{(t)}) - \nabla^2 \ell_j(\theta^{(j)}) \right\| + \left\| \sum_{j=0}^{t} \nabla^2 \ell_j(\theta^{(j)}) - \mathcal{I}\left(\theta^{(j)}\right) \right\|.$$

By Lemma C.1, $\nabla^2 \ell_j(\theta)$ is $K_3$ Lipschitz in $\theta$. Using this and the triangle inequality, the first term is bounded by

$$K_3 \sum_{j=1}^{t} \left\| \theta^{(t)} - \theta^{(j)} \right\| \leq K_3 \left( t \left\| \theta^{(t)} - \theta^{(0)} \right\| + \sum_{j=1}^{t-1} \left\| \theta^{(j)} - \theta^{(0)} \right\| \right) \leq 2K_3 t \max_{j \leq t} \left\| \theta^{(j)} - \theta^{(0)} \right\|.$$

By Lemma B.3, there exists an absolute constant $C > 0$ such that with probability at least $1 - \delta$ the second term is at most $CK_2 \sqrt{\frac{(t+1) \log\left(\frac{2d}{\delta}\right)}{n}}$, concluding the proof. $\square$

**Lemma C.3.** *Under Assumptions 1, 2, for any $t \in \mathbb{N}$, if $\theta^{(0)}, \ldots, \theta^{(t)} \in B_r(\theta^{(0)})$, then there exists an absolute constant $C > 0$ such that for any $\delta > 0$, with probability at least $1 - \delta$*

$$\left\| \sum_{j=0}^{t} \nabla^2 \ell_j(\theta^{(t)}) - (t+1)\mathcal{I}(\theta^{(0)}) \right\| \leq C(K_2 + K_3) \left( \sqrt{\frac{(t+1) \log\left(\frac{2d}{\delta}\right)}{n}} + t \max_{j \leq t} \left\| \theta^{(j)} - \theta^{(0)} \right\| \right).$$

*Proof.* By C.1, $\mathcal{I}(\theta)$ is $K_3$ Lipschitz in $\theta$, so

$$\left\| \sum_{j=0}^{t} \nabla^2 \ell_j(\theta^{(t)}) - (t+1)\mathcal{I}(\theta^{(0)}) \right\| \leq \left\| \sum_{j=0}^{t} \nabla^2 \ell_j(\theta^{(t)}) - \mathcal{I}(\theta^{(j)}) \right\| + \left\| (t+1)\mathcal{I}(\theta^{(0)}) - \sum_{j=0}^{t} \mathcal{I}(\theta^{(j)}) \right\|$$

$$\leq \left\| \sum_{j=0}^{t} \nabla^2 \ell_j(\theta^{(t)}) - \mathcal{I}(\theta^{(j)}) \right\| + K_3 \sum_{j=1}^{t} \left\| \theta^{(t)} - \theta^{(0)} \right\|$$

$$\leq \left\| \sum_{j=0}^{t} \nabla^2 \ell_j(\theta^{(t)}) - \mathcal{I}(\theta^{(j)}) \right\| + K_3 t \max_{j \leq t} \left\| \theta^{(j)} - \theta^{(0)} \right\|.$$

The proof now follows immediately from Lemma C.2. $\square$

We now prove the following proposition, which will serve a substantial role in the proof of Theorem 4.1.

**Proposition C.1.** *Under Assumptions 1 - 3, there exist constants $c := c(K_1, K_2, K_3, \lambda_0, r) > 0$ and $C := C(K_1, \lambda_0) > 0$ and a constant $C_2 := C_2(K_2, K_3)$ given by Lemma C.2 such that for any $t \in \mathbb{N}$, if $\max_{j \leq t-1} \left\| \theta^{(j)} - \theta^{(0)} \right\| \leq \max\left( \frac{\lambda_0}{4C_2}, r/2 \right)$, then for any $\delta > 0$, and $n \geq c \log\left(\frac{4d}{\delta}\right)$, with probability at least $1 - \delta$*

$$\left\| \theta^{(t)} - \theta^{(t-1)} \right\| \leq \max\left( \frac{C}{t} \sqrt{\frac{\log\left(\frac{4d}{\delta}\right)}{n}}, \frac{r}{2} \right).$$

*Proof.* Fix some $a > 0$ that will be specified later, and let $\mathbb{S}_a := \mathbb{S}_a(\theta^{(t-1)})$ be the sphere of radius $a$ with center at $\theta^{(t-1)}$. We will show that for sufficiently small $a$, with high probability it will hold simultaneously for all $\theta$ on the sphere $\mathbb{S}_a$ that $\sum_{j=0}^{t-1} \ell_j(\theta) > \sum_{j=0}^{t-1} \ell_j(\theta^{(t-1)})$. As a result, with high probability, there must be a local minimum of $\sum_{j=0}^{t-1} \ell_j(\theta)$ within the ball of radius $a$ centered at $\theta^{(t-1)}$. This implies[2] that $\left\| \theta^{(t+1)} - \theta^{(t)} \right\| \leq a$.

Assume for now that $a$ is small enough such that $\mathbb{S}_a \subseteq B_r(\theta^{(0)})$. We will later ensure this explicitly by picking $a < r/2$ (which is sufficient due to the assumption that $\left\| \theta^{(t-1)} - \theta^{(0)} \right\| \leq r/2$).

We first Taylor expand the normalized negative log-likelihood around $\theta^{(t-1)}$,

$$\sum_{j=0}^{t-1} \ell_j(\theta) - \ell_j(\theta^{(t-1)}) = \sum_{j=0}^{t-1} \nabla\ell_j(\theta^{(t-1)})^\top (\theta - \theta^{(t-1)}) + Q(\theta) + R(\theta), \tag{6}$$

where $Q(\theta)$ is the quadratic term, given by

$$Q(\theta) := \frac{1}{2} \sum_{j=0}^{t-1} (\theta - \theta^{(t-1)})^\top \nabla^2 \ell_j(\theta^{(t-1)})(\theta - \theta^{(t-1)}), \tag{7}$$

and $R(\theta)$ is the remainder term, which for some $\tilde{\theta}$ between $\theta$ and $\theta^{(t-1)}$ satisfies

$$|R(\theta)| = \frac{1}{6} \sum_{j=0}^{t-1} \sum_{i=1}^{d} \sum_{r=1}^{d} \sum_{k=1}^{d} \left( \frac{\partial^3}{\partial\theta_i \partial\theta_r \partial\theta_k} \ell_j(\tilde{\theta}) \right) (\theta - \theta^{(t-1)})_i (\theta - \theta^{(t-1)})_r (\theta - \theta^{(t-1)})_k$$

$$\leq \frac{1}{6} \sum_{j=0}^{t-1} \left\| \nabla^3 \ell_j(\tilde{\theta}) \right\| a^3 \leq \frac{tK_3}{6} a^3, \tag{8}$$

where the last inequality follows from 2. C, and by the convexity of $B_r(\theta^{(0)})$ which implies that $\tilde{\theta} \in B_r(\theta^{(0)})$.

For the linear term, first note that if $t \geq 2$ then $\theta^{(t-1)}$ is a stationary point of $\sum_{j=0}^{t-2} \ell_j(\cdot)$, so $\sum_{j=0}^{t-2} \nabla\ell_j(\theta^{(t-1)})^\top = 0$. So for any $t \in \mathbb{N}$, $\sum_{j=0}^{t-1} \nabla\ell_j(\theta^{(t-1)}) = \nabla\ell_{t-1}(\theta^{(t-1)})$. Using this and Lemma B.2, there exists a constant $C_1 := C_1(K_1) > 0$ such that with probability at least $1 - \delta/2$,

$$\left| \sum_{j=0}^{t-1} \nabla\ell_j(\theta^{(t-1)})^\top (\theta - \theta^{(t-1)}) \right| = \left| \nabla\ell_{t-1}(\theta^{(t-1)})^\top (\theta - \theta^{(t-1)}) \right|$$

$$\leq \left\| \nabla\ell_{t-1}(\theta^{(t-1)}) \right\| \left\| \theta - \theta^{(t-1)} \right\| \leq C_1 a \sqrt{\frac{\log\left(\frac{4d}{\delta}\right)}{n}}. \tag{9}$$

For the quadratic term, since the matrix $\nabla^2 \ell_j(\theta^{t-1})$ and the Fisher information matrices are symmetric, we have by Weyl's inequality, Assumption 3 and Lemma C.2 that for $C_2 = C_2(K_2, K_3) > 0$ it holds with probability at least $1 - \delta/2$ that

$$\lambda_{\min}\left( \sum_{j=0}^{t-1} \nabla^2 \ell_j(\theta^{(t-1)}) \right) \geq \lambda_{\min}\left( \sum_{j=0}^{t-1} \mathcal{I}(\theta^{(j)}) \right) - \left\| \sum_{j=0}^{t-1} \nabla^2 \ell_j(\theta^{(t-1)}) - \mathcal{I}(\theta^{(j)}) \right\|$$

$$\geq t\lambda_0 - C_2 \left( \sqrt{\frac{t \log\left(\frac{4d}{\delta}\right)}{n}} + t \max_{j \leq t-1} \left\| \theta^{(j)} - \theta^{(0)} \right\| \right) \tag{10}$$

---

[2] Here we use that if the argmax in the definition of MLE is not unique, it chooses the parameters closest to $\theta^{(t-1)}$. See the discussion following Def. (3.1) for more details.

Plugging Eq. (10) back into the quadratic term given by Eq. (7) and using the assumption that $\max_{j \leq t-1} \left\| \theta^{(j)} - \theta^{(0)} \right\| \leq \lambda_0/(4C_2)$ we have

$$Q \geq \frac{t}{2} \left( \lambda_0 - C_2 \max_{t \leq t-1} \left\| \theta^{(t)} - \theta^{(0)} \right\| - C_2 \sqrt{\frac{\log\left(\frac{4d}{\delta}\right)}{tn}} \right) a^2$$

$$\geq \frac{t}{2} \left( \frac{3}{4}\lambda_0 - C_2 \sqrt{\frac{\log\left(\frac{4d}{\delta}\right)}{tn}} \right) a^2, \tag{11}$$

where the last inequality follows by assumption.

Now take $a = \frac{8C_1}{t\lambda_0} \sqrt{\frac{\log\left(\frac{4d}{\delta}\right)}{n}}$. We can choose some constant $c := c\left(K_1, K_2, K_3, \lambda_0, r\right) > 0$ (independent of $t$) such that for any $n \geq c \log\left(\frac{4d}{\delta}\right)$, all of the following hold:

1. $a < \frac{r}{2}$,

2. $a < \frac{2C_1}{\sqrt{t}C_2}$,

3. $a < \frac{3\lambda_0}{4K_3}$.

The first condition was needed at the beginning of the proof. The second condition will allow us to bound Eq. (11), since together with the choice of $a$ it ensures that

$$C_2 \sqrt{\frac{\log\left(\frac{4d}{\delta}\right)}{tn}} = \frac{a\sqrt{t}C_2\lambda_0}{8C_1} < \frac{\lambda_0}{4}.$$

As a result, Eq. (11) becomes

$$Q > \frac{t}{2} \left( \frac{3}{4}\lambda_0 - \frac{\lambda_0}{4} \right) a^2 = \frac{t\lambda_0}{4} a^2. \tag{12}$$

The third condition on $a$ ensures that the remainder term from Eq. (8) is negligible, as

$$|R(\theta)| \leq \frac{tK_3}{6} a^3 < \frac{t\lambda_0}{8} a^2 < \frac{Q}{2}.$$

Notice that the choice of $a$ ensures that the bound for the linear term in Eq. (9) becomes

$$\left| \sum_{j=0}^{t-1} \nabla \ell_j(\theta^{(t-1)})^\top (\theta - \theta^{(t-1)}) \right| \leq C_1 a \sqrt{\frac{\log\left(\frac{4d}{\delta}\right)}{n}} = \frac{t\lambda_0}{8} a^2 < \frac{Q}{2}.$$

So overall, the Taylor expansion Eq. (6) satisfies

$$\sum_{j=0}^{t-1} \ell_j(\theta) - \ell_j(\theta^{(t-1)}) > -\frac{Q}{2} + Q - \frac{Q}{2} > 0.$$

So we have shown that for $a = \frac{8C_1}{t\lambda_0} \sqrt{\frac{\log\left(\frac{4d}{\delta}\right)}{n}}$ and $n \geq c \log\left(\frac{4d}{\delta}\right)$, it holds with probability at least $1 - \delta$ that for all $\theta \in \mathbb{S}_a$, $\sum_{j=0}^{t-1} \ell_j(\theta) > \sum_{j=0}^{t-1} \ell_j(\theta^{(t-1)})$. This implies the desired result as discussed at the beginning of the proof. $\qquad \square$

## C.2 Lemmas for Theorem 4.1

**Lemma C.4.** *Under Assumption 2, for any $t \in \mathbb{N}$, if there exists some open ball $B \subseteq \Theta$ such that $\theta^{(t)}, \theta^{(t+1)} \in B$, then there exists a matrix $R_t \in \mathbb{R}^{d \times d}$ with $\|R_t\| \leq \frac{t+1}{2} K_3 \left\| \theta^{(t+1)} - \theta^{(t)} \right\|$ such that*

$$\sum_{j=0}^{t} \nabla \ell_j \left( \theta^{(t+1)} \right) = \left( \sum_{j=0}^{t} \nabla \ell_j \left( \theta^{(t)} \right) + \nabla^2 \ell_j(\theta^{(t)}) \cdot (\theta^{(t+1)} - \theta^{(t)}) \right) + R_t(\theta^{(t+1)} - \theta^{(t)}).$$

*Proof.* Fix some coordinate $i \in [d]$ and consider the Taylor expansion of $\frac{\partial}{\partial \theta_i} \sum_{j=0}^{t} \ell_j$ around $\theta^{(t)}$, which gives that for some $\mathbf{z}_i \in \mathbb{R}^d$ that lies in the line segment between $\theta^{(t)}$ and $\theta^{(t+1)}$,

$$\frac{\partial}{\partial \theta_i} \sum_{j=0}^{t} \ell_j(\theta^{(t+1)}) = \frac{\partial}{\partial \theta_i} \sum_{j=0}^{t} \ell_j(\theta^{(t)}) + \sum_{k=1}^{d} \frac{\partial^2}{\partial \theta_k \partial \theta_i} \sum_{j=0}^{t} \ell_j(\theta^{(t)})(\theta^{(t+1)} - \theta^{(t)})_k$$

$$+ \frac{1}{2} \sum_{r=1}^{d} \sum_{k=1}^{d} \frac{\partial^3}{\partial \theta_r \partial \theta_k \partial \theta_i} \sum_{j=0}^{t} \ell_j(\mathbf{z}_i)(\theta^{(t+1)} - \theta^{(t)})_k (\theta^{(t+1)} - \theta^{(t)})_r, \quad (13)$$

where $\mathbf{z}_i \in B$ (and in particular, $\mathbf{z}_i \in \Theta$). Let $R_t \in \mathbb{R}^{d \times d}$ be the matrix whose coordinates are given by $[R_t]_{i,k} := \frac{1}{2} \sum_{j=0}^{t} \sum_{r=1}^{d} \frac{\partial^3}{\partial \theta_r \partial \theta_k \partial \theta_i} \ell_j(\mathbf{z}_i)(\theta^{(t+1)} - \theta^{(t)})_r$. Then Eq. (13) implies

$$\sum_{j=0}^{t} \nabla \ell_j \left( \theta^{(t+1)} \right) = \left( \sum_{j=0}^{t} \nabla \ell_j \left( \theta^{(t)} \right) + \nabla^2 \ell_j(\theta^{(t)}) \cdot (\theta^{(t+1)} - \theta^{(t)}) \right) + R_t(\theta^{(t+1)} - \theta^{(t)}).$$

It remains to bound $\|R_t\|$. By Assumption 2. C, we have

$$\|R_t\| = \sup_{\mathbf{v}_1, \mathbf{v}_2 \neq 0} \mathbf{v}_1^T R_t \mathbf{v}_2 = \frac{1}{2} \sum_{j=0}^{t} \nabla^3 \ell_j(\mathbf{z}_i) \left( \mathbf{v}_1, \mathbf{v}_2, \theta^{(t+1)} - \theta^{(t)} \right)$$

$$\leq \frac{t+1}{2} K_3 \|\mathbf{v}_1\| \|\mathbf{v}_2\| \left\| \theta^{(t+1)} - \theta^{(t)} \right\|,$$

which shows $\|R_t\| \leq \frac{t+1}{2} K_3 \left\| \theta^{(t+1)} - \theta^{(t)} \right\|$. $\qquad\square$

**Lemma C.5.** *Let $A, B \in \mathbb{R}^{d \times d}$ be positive definite matrices, then*

$$\left\| A^{-1} - B^{-1} \right\| \leq \frac{\|A - B\|}{\lambda_{\min}(A)\lambda_{\min}(B)}$$

*Proof.*

$$\left\| A^{-1} - B^{-1} \right\| = \left\| A^{-1}(B - A)B^{-1} \right\| \leq \left\| A^{-1} \right\| \|A - B\| \left\| B^{-1} \right\| = \frac{\|A - B\|}{\lambda_{\min}(A)\lambda_{\min}(B)}.$$

$\qquad\square$

# D  Proof of Theorem 4.1

**Theorem 4.1.** *Under Assumptions 1 - 3, there exist constants $c, C > 0$ which depend only on $K_1, K_2, K_3, \lambda_0$ and $r$, such that for any $T \in \mathbb{N}$, $\delta > 0$ and any $n \geq c \left( \log(T) + 1 \right)^2 \log^2 \left( \frac{7dT}{\delta} \right)$, it holds with probability at least $1 - \delta$ that*

$$\left\| \theta^{(T)} - \theta^\star \right\| \leq C \sqrt{\frac{\log\left(\frac{4d}{\delta}\right)}{n}}. \tag{1}$$

*Proof.* Let $C_1 := C_1(K_1, K_2, K_3, \lambda_0, r) > 0$ denote the maximum of the constants appearing in the statements of Lemmas B.1, B.2, B.3, B.4, C.2, C.3 and Proposition C.1, and let $\delta_0, \ldots, \delta_T > 0$ be given by $\delta_t := \delta/(2T)$ for $t < T$ and $\delta_T = \delta/2$. Let $C$ and $c$ be constants as in the theorem statement, whose values will be determined throughout the proof, and set

$$N := \frac{c}{49} \left( \log(T) + 1 \right)^2 \log \left( \frac{24dT}{\delta_0} \right)^2 \leq c \left( \log(T) + 1 \right)^2 \log \left( \frac{7dT}{\delta} \right)^2. \tag{14}$$

We will show inductively on $t = 0, \ldots, T$ that for any $n \geq N$, it holds with probability at least $1 - \frac{1}{2} t \delta_0 - \frac{1}{2} \sum_{j=1}^{t} \delta_j$ that

$$\left\| \theta^{(\tau)} - \theta^{(0)} \right\| \leq \min \left( C \sqrt{\frac{\log\left(\frac{2d}{\delta_\tau}\right)}{n}}, \frac{r}{2} \right), \qquad \forall \tau \in \{0, \ldots, t\}. \tag{15}$$

Note that in the case of $t = T$, the probability of Eq. (15) holding becomes $1 - \frac{1}{2}T\delta_0 - \frac{1}{2}\sum_{j=1}^{T}\delta_j \geq 1 - \delta$ and the theorem follows.

For $t = 0$ the claim is trivial. Now, assume Eq. (15) holds for $t - 1$, and we will prove it holds for $t$.

By Eq. (15), for sufficiently large $c$ and the assumption that $n \geq N$, the conditions of Proposition C.1 are satisfied (if Eq. (15) is not $< \frac{\lambda_0}{4C_2}$ as Proposition C.1 requires, one can replace $c$ by a suitable larger constant that depends on the same parameters), so it implies that with probability at least $1 - \delta_0/6$ (using the union bound),

$$\left\| \theta^{(\tau+1)} - \theta^{(\tau)} \right\| \leq \max \left( \frac{C_1}{\tau+1}\sqrt{\frac{\log\left(\frac{24dt}{\delta_0}\right)}{n}}, \frac{r}{2} \right), \qquad \forall \tau \in \{0, \ldots, t-1\}. \quad (16)$$

We let $A_1$ denote the event that Eq. (15) and Eq. (16) indeed hold. By the union bound, $P(A_1) \geq 1 - \frac{1}{2}(t-1)\delta_0 - \frac{1}{2}\sum_{j=1}^{t-1}\delta_j - \delta_0/6$.

Consider some $\tau \in \{0, \ldots, t-1\}$. $\theta^{(\tau+1)}$ is defined as the MLE on $X^{(\leq \tau)}$, which in particular means that it is a stationary point of the log-likelihood function, so $\sum_{j=0}^{\tau} \nabla \ell_j(\theta^{(\tau+1)}) = 0$. When $A_1$ occurs, the conditions of Lemma C.4 are satisfied, which gives us a Taylor expansion for $\sum_{j=0}^{\tau} \ell_j(\theta^{(\tau+1)})$ as

$$0 = \sum_{j=0}^{\tau} \nabla\ell_j\left(\theta^{(\tau+1)}\right) = \left(\sum_{j=0}^{\tau}\nabla\ell_j\left(\theta^{(\tau)}\right) + \nabla^2\ell_j(\theta^{(\tau)}) \cdot (\theta^{(\tau+1)} - \theta^{(\tau)})\right) + R_\tau(\theta^{(\tau+1)} - \theta^{(\tau)}), \quad (17)$$

where $R_\tau$ is a matrix that satisfies by Eq. (16)

$$\|R_\tau\| \leq \frac{(\tau+1)K_3}{2}\left\|\theta^{(\tau+1)} - \theta^{(\tau)}\right\| \leq \frac{C_1 K_3}{2}\sqrt{\frac{\log\left(\frac{24dt}{\delta_0}\right)}{n}} \leq \frac{\lambda_0}{2}, \quad (18)$$

(where again the last inequality assumes $c$ is sufficiently large; if not, increase it).

By definition, for any $\tau > 0$, $\theta^{(\tau)}$ is the MLE for $X^{(\leq \tau-1)}$, so it is a stationary point satisfying $\sum_{j=0}^{\tau-1}\nabla\ell_j\left(\theta^{(\tau)}\right) = 0$. For notational simplicity, let $H_\tau := \left(\sum_{j=0}^{\tau}\nabla^2\ell_j(\theta^{(\tau)})\right) + R_\tau$, then Eq. (17) simplifies to

$$0 = \nabla\ell_\tau\left(\theta^{(\tau)}\right) + H_\tau(\theta^{(\tau+1)} - \theta^{(\tau)}). \quad (19)$$

To isolate $\theta^{(\tau+1)} - \theta^{(\tau)}$ we first show that $H_\tau$ is invertible. By Lemma C.3, with probability at least $1 - \delta_0/(6t)$,

$$\left\|\sum_{j=0}^{\tau}\nabla^2\ell_j(\theta^{(\tau)}) - (\tau+1)\mathcal{I}\left(\theta^{(0)}\right)\right\| \leq C_1\left(\sqrt{\frac{\log\left(\frac{24dt}{\delta_0}\right)}{n}} + \tau\max_{j\leq\tau}\left\|\theta^{(j)} - \theta^{(0)}\right\|\right)$$

$$\leq C_1\left(\sqrt{\frac{\log\left(\frac{24dt}{\delta_0}\right)}{n}} + \tau C\sqrt{\frac{\log\left(\frac{2d}{\delta_\tau}\right)}{n}}\right)$$

$$\leq (C_1 + C)\tau\sqrt{\frac{\log\left(\frac{24dt}{\delta_0}\right)}{n}}, \quad (20)$$

where the second inequality follows from Eq. (15) and that $\delta_t = \delta_0$ for $\tau < T$. Let $A_2$ denote the even that Eq. (20) is indeed satisfied for all $\tau \in \{0, \ldots, t - 1\}$, which by the union bound satisfies $P(A_2) \geq 1 - \delta_0/6$. When both $A_1$ and $A_2$ occur, using Weyl's inequality, Eq. (20) and Eq. (18) we have,

$$\lambda_{\min}\left(H_\tau\right) \geq \lambda_{\min}\left((\tau + 1)\mathcal{I}\left(\theta^{(0)}\right)\right) - \left\|\sum_{j=0}^{\tau} \nabla^2 \ell_j(\theta^{(\tau)}) - (\tau + 1)\mathcal{I}\left(\theta^{(0)}\right)\right\| - \|R_\tau\|$$

$$\geq (\tau + 1)\left(\frac{\lambda_0}{2} - (C_1 + C)\sqrt{\frac{\log\left(\frac{24dt}{\delta_0}\right)}{n}}\right) \geq (\tau + 1)\frac{\lambda_0}{4}, \tag{21}$$

where the last inequality follows for sufficiently large $c$ and the condition that $n \geq N$. In particular, under these events, every $H_\tau$ is invertible so Eq. (19) implies

$$\theta^{(\tau+1)} - \theta^{(\tau)} = -H_\tau^{-1}\nabla\ell_\tau\left(\theta^{(\tau)}\right).$$

Taking a telescopic sum, we obtain

$$\left\|\theta^{(t)} - \theta^{(0)}\right\| = \left\|\sum_{\tau=0}^{t-1} \theta^{(\tau+1)} - \theta^{(\tau)}\right\| = \left\|\sum_{\tau=0}^{t-1} H_\tau^{-1}\nabla\ell_\tau\left(\theta^{(\tau)}\right)\right\|$$

$$\leq \left\|\sum_{\tau=0}^{t-1} \frac{1}{\tau+1}\mathcal{I}(\theta^{(0)})^{-1}\nabla\ell_\tau\left(\theta^{(\tau)}\right)\right\| + \left\|\sum_{\tau=0}^{t-1}\left(H_\tau^{-1} - \frac{1}{\tau+1}\mathcal{I}(\theta^{(0)})^{-1}\right)\nabla\ell_\tau\left(\theta^{(\tau)}\right)\right\|$$

$$\leq \frac{1}{\lambda_0}\left\|\sum_{\tau=0}^{t-1} \frac{1}{\tau+1}\nabla\ell_\tau\left(\theta^{(\tau)}\right)\right\| + \sum_{\tau=0}^{t-1}\left\|H_\tau^{-1} - \frac{1}{\tau+1}\mathcal{I}(\theta^{(0)})^{-1}\right\| \cdot \left\|\nabla\ell_\tau\left(\theta^{(\tau)}\right)\right\|. \tag{22}$$

It remains to bound the terms in Eq. (22). We will first employ an additional probabilistic bound for the gradient terms. By Lemma B.1, with probability at least $1 - \delta_t/2$

$$\left\|\sum_{\tau=0}^{t-1} \frac{1}{\tau+1}\nabla\ell_\tau\left(\theta^{(\tau)}\right)\right\| \leq C_1\sqrt{\frac{\log\left(\frac{4d}{\delta_t}\right)}{n}}. \tag{23}$$

Similarly, by Lemma B.2 and the union bound, it holds with probability at least $1 - \delta_0/6$ that

$$\left\|\nabla\ell_\tau\left(\theta^{(\tau)}\right)\right\| \leq C_1\sqrt{\frac{\log\left(\frac{12dt}{\delta_0}\right)}{n}}, \qquad \forall \tau \in \{0, \ldots, t - 1\}. \tag{24}$$

Let $A_3$ denote the event that Eq. (23) and Eq. (24) are satisfied. Then letting $A := A_1 \cap A_2 \cap A_3$ be the intersection of the desired events in this proof, we have $\mathbb{P}(A) \geq 1 - \frac{1}{2}t\delta_0 - \frac{1}{2}\sum_{j=1}^{t}\delta_j$ as desired.

Under the event $A$, from Eq. (18, 20, 21) and Lemma C.5, it holds for all $\tau \in \{0, \ldots, t - 1\}$ that

$$\left\|H_\tau^{-1} - \frac{1}{\tau+1}\mathcal{I}(\theta^{(0)})^{-1}\right\| \leq \frac{\left\|H_\tau - (\tau + 1)\mathcal{I}(\theta^{(0)})\right\|}{\lambda_{\min}\left((\tau + 1)\mathcal{I}(\theta^{(0)})\right)\lambda_{\min}\left(H_\tau\right)}$$

$$\leq \frac{\left\|\sum_{j=0}^{\tau}\nabla^2\ell_j(\theta^{(\tau)}) - (\tau + 1)\mathcal{I}\left(\theta^{(0)}\right)\right\| + \|R_t\|}{(\tau + 1)\lambda_0\lambda_{\min}\left(H_\tau\right)}$$

$$\leq \frac{4}{(\tau + 1)\lambda_0^2}\left(C_1 + C + \frac{C_1K_3}{2}\right)\sqrt{\frac{\log\left(\frac{24dt}{\delta_0}\right)}{n}}. \tag{25}$$

Combining Eq. (24), Eq. (25) and the fact that $\sum_{\tau=1}^{t} \frac{1}{\tau} \leq 1 + \int_1^t \frac{1}{x} dx \leq 1 + \log(t)$, we have for a suitable $C' = C'(K_1, K_2, K_3, \lambda_0, r)$

$$\sum_{\tau=0}^{t-1} \left\| H_\tau^{-1} - \frac{1}{\tau+1} \mathcal{I}(\theta^{(0)})^{-1} \right\| \cdot \left\| \nabla \ell_\tau \left( \theta^{(\tau)} \right) \right\| \leq \sum_{\tau=0}^{t-1} \frac{C' \log \left( \frac{24 dt}{\delta_0} \right)}{n(\tau+1)}$$

$$\leq \sum_{\tau=0}^{t-1} \frac{C' \log \left( \frac{24 dt}{\delta_0} \right)}{n} \sum_{\tau=1}^{t} \frac{1}{\tau}$$

$$\leq \frac{1}{\sqrt{n}} \cdot \frac{C' \log \left( \frac{24 dt}{\delta_0} \right) (\log(t) + 1)}{\sqrt{n}}$$

$$\leq_{(\star)} \sqrt{\frac{1}{n}},$$

where $(\star)$ follows whenever $\sqrt{c} \geq C'$ by the assumption that

$$n \geq N \geq c \left( (\log(T) + 1) \log \left( \frac{24 dT}{\delta_0} \right) \right)^2.$$

Using this and Eq. (23), Eq. (22) reduces to

$$\left\| \theta^{(t)} - \theta^{(0)} \right\| \leq \left( \frac{C_1}{\lambda_0} + 1 \right) \sqrt{\frac{\log \left( \frac{2d}{\delta_t} \right)}{n}}.$$

Taking a suitable $C$ gives the desired bound. Lastly, for the induction we also need $\left\| \theta^{(t)} - \theta^{(0)} \right\| \leq \frac{r}{2}$. This is indeed the case, taking sufficiently large $c$.

$\square$

# E   Proof of Theorem 5.1

**Construction 1.** *Consider a fixed $N \in \mathbb{N}$ and let*

$$f(\alpha) := \begin{cases} \frac{1}{39} & \alpha \leq \frac{1}{10} \\ \frac{1}{32 \cdot (128)^{2N} - 1} & \alpha > \frac{1}{10} \end{cases}, \qquad \forall \alpha \in \mathbb{R}. \tag{26}$$

*Let $\mathcal{X} = \mathbb{R}$, $\Theta = \left\{ (\alpha, \mu) \mid \alpha \in \left[0, \frac{1}{4}\right] \mu \in [2, 3 - f(\alpha)] \right\}$. Letting $U$ denote the uniform distribution, we define the family of distributions given by:*

$$\frac{1}{2} U([0,1]) + \frac{1-\alpha}{2} U([0, 1-2\alpha]) + \frac{\alpha}{4} \left( U([2,3]) + U([\mu, \mu + f(\alpha)]) \right).$$

*Equivalently, letting $\mathbb{I}$ denote the indicator function (where for any set $A$, $\mathbb{I}_A(x)$ is 1 if $x \in A$ and 0 otherwise), the PDFs $p_\theta$ are given by:*

$$p_\theta(x) = \frac{1}{2} \mathbb{I}_{[0,1]}(x) + \frac{1-\alpha}{2(1-2\alpha)} \mathbb{I}_{[0,1-2\alpha]}(x) + \frac{\alpha}{4} \left( \mathbb{I}_{[2,3]}(x) + \frac{1}{f(\alpha)} \mathbb{I}_{[\mu, \mu+f(\alpha)]}(x) \right)$$

$$= \left( \frac{1}{2} + \frac{1-\alpha}{2(1-2\alpha)} \right) \mathbb{I}_{[0,1-2\alpha]}(x) + \frac{1}{2} \mathbb{I}_{[1-2\alpha,1]}(x)$$

$$+ \frac{\alpha}{4} \left( 1 + \frac{1}{f(\alpha)} \right) \mathbb{I}_{[\mu,\mu+f(\alpha)]}(x) + \frac{\alpha}{4} \mathbb{I}_{[2,3] \setminus [\mu,\mu+f(\alpha)]}(x).$$

*As such,*

$$-\log p_\theta(x) = -\log \left( \frac{1}{2} + \frac{1-\alpha}{2(1-2\alpha)} \right) \mathbb{I}_{[0,1-2\alpha]}(x) - \log \left( \frac{1}{2} \right) \mathbb{I}_{[1-2\alpha,1]}(x)$$

$$- \log \left( \frac{\alpha}{4} \left( 1 + \frac{1}{f(\alpha)} \right) \right) \mathbb{I}_{[\mu,\mu+f(\alpha)]}(x) - \log \left( \frac{\alpha}{4} \right) \mathbb{I}_{[2,3] \setminus [\mu,\mu+f(\alpha)]}(x). \tag{27}$$

**Lemma E.1.** *Under Construction 1, $P_\Theta$ is a TV-consistent family of distributions and $\theta^{(t)}$ exist.*

*Proof.* Consider some dataset $X \subseteq \mathcal{X}$ of size $k \in \mathbb{N}$, there is a finite number of values that $p_\theta(x)$ can take, depending on the interval $x$ lies in. This means,

$$|\{(p_\theta(x_1), \ldots, p_\theta(x_k)) \mid \theta \in \Theta\}| < \infty.$$

As such, there must be some $\theta$ that achieves this maximum.

Consistency of the MLEs follows from Lemma F.10 $\qquad\qquad\qquad\qquad\qquad\qquad\qquad\qquad$ $\square$

**Theorem 5.1.** *There exists $\Theta \subseteq \mathbb{R}^2$ and $\theta^\star \in \Theta$, such that for any $n \in \mathbb{N}$, there is a TV-consistent family of distributions $\{p_\theta\}_{\theta \in \Theta}$ (that may depend on $n$) such that*

1. *with probability at least $1 - \frac{1}{n}$,*

$$\mathrm{TV}\left(p_{\theta^\star}, p_{\theta^{(1)}}\right) \leq \frac{\log(n)}{n} .$$

2. *For some absolute constants $c, C > 0$, it holds with probability at least $c$ that*

$$\mathrm{TV}\left(p_{\theta^\star}, p_{\theta^{(2)}}\right) \geq C .$$

*Proof.* Consider the setting given by Construction 1 with $N = n$ and let $\theta^{(0)} = (\alpha^{(0)} = 0, \mu^{(0)} = 2)$. Existence of $\theta^{(1)}$ and $\theta^{(2)}$ as well as TV-consistency of $P_\Theta$ are given by Lemma E.1.

Because $\alpha^{(0)} = 0$, $p_{\theta^{(0)}}$ is supported on $[0, 1]$, meaning that $x_i^{(0)} \in [0, 1]$ for every $i \in [n]$. As such,

$$
\begin{aligned}
\ell_0(\theta) &= -\frac{1}{n} \sum_{i=1}^{n} \log\left(p_\theta(x_i^{(0)})\right) \\
&= -\log\left(\frac{1}{2} + \frac{1-\alpha}{2(1-2\alpha)}\right) \frac{1}{n} \sum_{i=1}^{n} \mathbb{I}_{[0, 1-2\alpha]}(x_i^{(0)}) - \log\left(\frac{1}{2}\right) \frac{1}{n} \sum_{i=1}^{n} \mathbb{I}_{[1-2\alpha, 1]}(x_i^{(0)}) \\
&= -\log\left(\frac{1}{2}\right) - \log\left(1 + \frac{1-\alpha}{1-2\alpha}\right) \frac{1}{n} \sum_{i=1}^{n} \mathbb{I}_{[0, 1-2\alpha]}(x_i^{(0)}),
\end{aligned}
$$

where the last equality used $\log\left(\frac{1}{2} + \frac{1-\alpha}{2(1-2\alpha)}\right) = \log\left(\frac{1}{2}\left(1 + \frac{1-\alpha}{1-2\alpha}\right)\right) = \log\left(\frac{1}{2}\right) + \left(1 + \frac{1-\alpha}{1-2\alpha}\right)$, and that $x_i^{(0)} \in [0, 1]$.

Let $x_{\max} := \max_{i \in [n]} x_i^{(0)}$. Note that whenever $\alpha \leq \frac{1-x_{\max}}{2}$, then every $x_i^{(0)}$ is inside the interval $[0, 1-2\alpha]$. Consequently, for all $\alpha \in \left[0, \frac{1-x_{\max}}{2}\right]$, $\ell_0(\theta) = -\log\left(\frac{1}{2}\right) - \log\left(1 + \frac{1-\alpha}{1-2\alpha}\right)$. Since the function $-\log\left(1 + \frac{1-\alpha}{1-2\alpha}\right)$ is monotonically decreasing in $\alpha$ for all $\alpha < \frac{1}{2}$, $\ell_0(\theta)$ is also monotonically decreasing on $\left[0, \frac{1-x_{\max}}{2}\right]$. As such, the MLE $\theta^{(1)} = (\alpha^{(1)}, \mu^{(1)})$ which minimizes $\ell_0(\theta)$ must satisfy

$$\alpha^{(1)} = \frac{1 - x_{\max}}{2}. \tag{28}$$

**Consistency of $\theta^{(1)}$:** By Eq. (28) and Lemma F.8 for any $\delta > 0$, it holds with probability at least $1 - \delta$ that

$$\alpha^{(1)} = \frac{1 - x_{\max}}{2} \leq \frac{\log\left(\frac{1}{\delta}\right)}{2n}.$$

Now using this and that $p_{\theta^{(0)}} = \mathbb{I}_{[0,1]}(x)$, the total variation can be bounded as

$$
\begin{aligned}
\mathrm{TV}\left(p_{\theta^{(0)}}, p_{\theta^{(1)}}\right) =& \frac{1}{2}\int_0^1 |1 - p_{\theta^{(1)}}|(x)dx + \frac{1}{2}\int_2^3 p_{\theta^{(1)}}(x)dx \\
=& \frac{1}{2}\int_0^{1-2\alpha^{(1)}}\left|1 - \left(\frac{1}{2} + \frac{1-\alpha^{(1)}}{2(1-2\alpha^{(1)})}\right)\right|dx + \frac{1}{2}\int_{1-2\alpha^{(1)}}^1 \left|1 - \frac{1}{2}\right|dx + \frac{\alpha^{(1)}}{4} \\
=& \frac{1-2\alpha^{(1)}}{2}\cdot\left|\frac{1}{2} - \frac{1-\alpha^{(1)}}{2(1-2\alpha^{(1)})}\right| + \frac{3}{4}\alpha^{(1)} \\
=& \frac{\alpha^{(1)}}{4} + \frac{3}{4}\alpha^{(1)} \leq \frac{\log\left(\frac{1}{\delta}\right)}{2n}.
\end{aligned}
$$

**Inconsistency of $\theta^{(2)}$:** We will now show that with some constant probability, there will be some $x_i^{(1)} \in [2,3]$. Let $A$ denote the event that $x_{\max} \leq 1 - \frac{1}{n}$. Since $x_i^{(0)} \sim U([0,1])$ i.i.d, we have

$$
P(A) = \left(1 - \frac{1}{n}\right)^n \leq \frac{1}{e}.
$$

Conditioned on $A$, we have $\alpha^{(1)} \geq \frac{1-x_{\max}}{2} \geq \frac{1}{2n}$, so for each $x_i^{(1)} \sim p_{\theta^{(1)}}$,

$$
\mathbb{P}\left(x_i^{(1)} \in [2,3] \mid A\right) \geq \frac{\alpha^{(1)}}{4} \geq \frac{1}{8n}.
$$

Therefore, the probability that none of the $x_i^{(1)}$ fall in $[2,3]$ is at most

$$
\mathbb{P}\left(\forall i \in [n],\ x_i^{(1)} \notin [2,3] \mid A\right) \leq \left(1 - \frac{1}{8n}\right)^n \leq e^{-1/8}.
$$

Applying the law of total probability,

$$
\mathbb{P}\left(\exists i \in [n] \text{ such that } x_i^{(1)} \in [2,3]\right) \geq \mathbb{P}(A)\cdot\mathbb{P}\left(\exists i,\ x_i^{(1)} \in [2,3] \mid A\right) \geq \frac{1}{e}\cdot(1 - e^{-1/8}).
$$

Thus, with constant probability, one of the samples $x_i^{(1)}$ lies in $[2,3]$. The remainder of the proof is conditioned on this occurring. We will now show that the existence of $x_i^{(1)} \in [2,3]$ implies that $\alpha^{(2)}$ will be far from $\alpha^{(0)} = 0$.

Now consider any $\alpha \in [0,1/10]$. The function $f$ (defined in Eq. (26)) satisfies $f(\alpha) = \frac{1}{39}$ for any such $\alpha$. As such, the term $\frac{\alpha}{4}\left(1 + \frac{1}{f(\alpha)}\right)$ is at most 1 for any $\alpha \in [0,1/10]$. Consequently, for any $\alpha \in [0,1/10]$, the only term in Eq. (27) that is negative is the first one, meaning for any $x$ we have

$$
-\log p_\theta(x) \geq -\log\left(\frac{1}{2} + \frac{1-\alpha}{2(1-2\alpha)}\right).
$$

Since this bound is monotonically decreasing in $\alpha$, we have for any $\alpha \in [0,1/10]$,

$$
\sum_{t=0}^1 \ell_t(\theta) = -\frac{1}{n}\sum_{t=0}^1\sum_{i=1}^n \log p_\theta(x_i^{(t)}) \geq -2\log\left(\frac{1}{2} + \frac{1-\alpha}{2(1-2\alpha)}\right) \geq -2\log\left(\frac{17}{16}\right).
$$

Now let $\bar{\alpha} = 1/8$ and fix $\bar{\mu}$ such that there is at least one sample in $[\bar{\mu}, \bar{\mu} + f(\bar{\alpha})]$ (which we know exists as there is some $x_i^{(1)} \in [2,3]$). Plugging this $\bar{\theta} = (\bar{\alpha}, \bar{\mu})$ into Eq. (27), using that the first term

is negative, and that there is at least one $x_i^{(1)} \in [2,3]$, we have:

$$\sum_{t=0}^{1} \ell_t\left(\bar{\theta}\right) \leq -2\log\left(\frac{1}{2}\right) - 2\log\left(\frac{1}{32}\right) - \frac{1}{n}\log\left(\frac{1}{32}\left(1 + \frac{1}{f(\frac{1}{8})}\right)\right)$$

$$= -2\log\left(\frac{1}{64}\right) - \frac{1}{n}\log\left(\frac{1}{32}\left(1 + 32 \cdot (128)^{2N} - 1\right)\right)$$

$$= -2\log\left(\frac{1}{64}\right) - 2\frac{N}{n}\log(128) \leq -2\log(2)$$

$$\leq \inf_{\theta \,:\, \alpha \leq 1/10} \sum_{t=0}^{1} \ell_t\left(\theta\right)$$

As such, we have shown that $\alpha^{(2)} \notin [0, 1/10]$. As a result, the TV distance can be lower bounded as

$$\mathrm{TV}\left(p_{\theta^{(0)}}, p_{\theta^{(2)}}\right) \geq \int_{0}^{1-2\alpha^{(2)}} |p_{\theta^{(0)}} - p_{\theta^{(2)}}| = \left|1 - \frac{1}{2} - \frac{1-\alpha^{(2)}}{2(1-2\alpha^{(2)})}\right|$$

$$= \left|\frac{1}{2}\left(1 - \frac{1-\alpha^{(2)}}{1-2\alpha^{(2)}}\right)\right| \geq \frac{1}{16}.$$

$\square$

# F    Proof of Theorem 5.2

**Construction 2.** *Let*

$$A := \left\{(\alpha_j)_{j=0}^{\infty} \in [0,\,1/4]^{\mathbb{N} \cup \{0\}} \mid \exists\, j_* \in \mathbb{N} \text{ s.t } \forall j \geq j_*, \ \alpha_j = 0\right\},$$

*namely, the set of all countable tuples in $[0, 1/4]^{\mathbb{N} \cup \{0\}}$ that have a finite number of non zero entries. For any $\boldsymbol{\alpha} \in A$, let*

$$h_{\boldsymbol{\alpha}}(x) := \sum_{j=0}^{\infty} (1-\alpha_j)\left(\prod_{k=0}^{j-1} \alpha_k\right)\frac{1}{1-2\alpha_j}\mathbb{I}_{[j, j+1-2\alpha_j]}(x),$$

*where we use the notational convention that $\prod_{k=0}^{-1} \alpha_k = 1$.*

*To see that this is a valid PDF, first note that $\int_{-\infty}^{\infty} h_{\boldsymbol{\alpha}}(x)dx = \sum_{j=0}^{\infty}(1-\alpha_j)\left(\prod_{k=0}^{j-1} \alpha_k\right)$. Now consider any fixed $M \in \mathbb{N}$, then*

$$\sum_{j=0}^{M}(1-\alpha_j)\left(\prod_{k=0}^{j-1}\alpha_k\right) = \sum_{j=0}^{M}\left(\prod_{k=0}^{j-1}\alpha_k - \prod_{k=0}^{j}\alpha_k\right) = 1 - \prod_{k=0}^{M}\alpha_k.$$

*In particular, since $\alpha_k \in [0, 1/4]$, this converges to 1 as $M \to \infty$.*

*Let $f : [2, \infty) \to (0, 1/2)$ be a monotonically decreasing function that will be specified later in the proof. We also define for any $\beta \in [0, 1]$ and $J \in \mathbb{N}$,*

$$g_{\beta,J}(x) := \frac{1}{2J}\mathbb{I}_{[0,J]}(x) + \frac{1}{2f(J)}\mathbb{I}_{[J-\beta,J-\beta+f(J)]}(x).$$

*The parameters $\theta$ will consist of tuples $(\boldsymbol{\alpha}, \beta, J, s)$ where $s \in \{0, 1\}$ is a "selector" which tells us if we should choose the PDF $h_{\boldsymbol{\alpha}}$ or the PDF $g_{\beta,J}$. Specifically, the parameter space is $\Theta = A \times [0, 1] \times (\mathbb{N} \setminus \{1\}) \times \{0, 1\}$. And the distributions $P_{\Theta}$ are given by*

$$p_{\theta}(x) = \begin{cases} h_{\boldsymbol{\alpha}}(x) & s = 0 \\ g_{\beta,J}(x) & s = 1 \end{cases}.$$

*Consider the ground truth distribution $\theta^{(0)}$ to be such that*

$$p_{\theta^{(0)}}(x) = h_\mathbf{0}(x) = \mathbb{I}_{[0,1]}(x).$$

*For each $t$, $\theta^{(t)}$ is an MLE given the data $X^{(\leq t)}$. Existence will be guaranteed in Lemma F.1. Regarding uniqueness, we do not use the fact that $\theta^{(t)}$ is the closest maximizer of the log likelihood to $\theta^{(t-1)}$. This is completely unimportant to the proof.*

*Lastly, for convenience, let*

$$M_{t,j} := \begin{cases} \max\left(X^{(\leq t)} \cap [j, j+1]\right) - j & \exists x \in X^{(\leq t)} \cap [j, j+1] \\ 0 & else \end{cases},$$

*where the maximum exists because the set is finite. In words, $M_{t,j}$ denotes the maximal observed offset within the $j$'th interval $[j, j+1]$ up to time $t$.*

**Remark 1.** *Under Construction 2, for any $x \in \mathcal{X}$, if there is some non-negative integer $j(x)$ such that $x \in [j(x), j(x) + 1 - 2\alpha_{j(x)}]$, then*

$$\log\left(h_{\boldsymbol{\alpha}}(x)\right) = \log\left(\frac{1 - \alpha_{j(x)}}{1 - 2\alpha_{j(x)}}\left(\prod_{k=0}^{j(x)-1} \alpha_k\right)\right) = \log\left(1 + \frac{\alpha_{j(x)}}{1 - 2\alpha_{j(x)}}\right) + \sum_{k=0}^{j(x)-1} \log\left(\alpha_k\right). \tag{29}$$

*If no such $j(x)$ exists, then $h_{\boldsymbol{\alpha}}(x) = 0$ and $\log\left(h_{\boldsymbol{\alpha}}(x)\right)$ is undefined.*

**Lemma F.1.** *Under Construction 2, $P_\Theta$ is a TV-consistent family of distributions and $\theta^{(t)}$ exist.*

*Proof.* Consider some dataset $X$ of size $k$ and fix $b \in \mathbb{N}$ such that $X \subseteq [0, b]$ Following Remark 1 as well as the definition of $g_{\beta,J}$, it is straightforward to see that for any $x_i$, there is a finite number of values that $p_\theta(x)$ can take, depending on the interval $x$ lies in. This means,

$$|\{(p_\theta(x_1), \ldots, p_\theta(x_n)) \mid \theta \in \Theta\}| < \infty.$$

As such, there must be some $\theta$ that achieves this maximum.

Note that any PDF in $P_\Theta$ has finite support. So w.l.o.g we may assume that $\text{supp}(p_{\theta^{(\star)}}) \subseteq [0, b]$ so that for any $n$, samples $x_1, \ldots, x_n$ from $p_{\theta^{(\star)}}$ will all be in $[0, b]$.

By Remark 1, for any $\theta \in \Theta$ and $j \geq b$, the parameters $\alpha_j$ do not affect the log likelihood. Furthermore, since $g_{\beta,J} = \frac{1}{2J}\mathbb{I}_{[0,J]}(x)$ the log likelihood is strictly decreasing in $J$ for $\forall J \geq b+1$. As such, for the purpose of showing TV-consistency, we may "discard" all values of $J \geq b+1$ and all indices $\geq J+1$ in $\boldsymbol{\alpha}$, treating $\Theta$ as $[0, \frac{1}{4}]^{J+1} \times [0, 1] \times 2, \ldots, J+1 \times \{0, 1\}$. This is a closed and bounded subset of a Euclidean space and is therefore compact. Furthermore, $\log p_{\theta^{(0)}}(x)$ are uniformly bounded. So by Lemma F.10, the MLE is consistent. $\square$

**Lemma F.2.** *For any $t \in \mathbb{N} \cup \{0\}$ with $s^{(t)} = 0$, and any $j \in \mathbb{N} \cup \{0\}$, if $M_{t,j} > 0$ then*

$$\alpha_j^{(t+1)} = \frac{1}{2}\left(1 - M_{t,j}\right).$$

*Proof.* This is a direct consequence of Remark 1. Specifically, from Eq. (29) it follows that the log likelihood is strictly increasing in $\alpha_j^{(t+1)}$, and is subject to the constraint that for all $x \in [j, j+1]$ it holds that $x \in [j, j+1 - 2\alpha_j^{(t+1)}]$. In particular, this implies that any maximizer must satisfy

$$M_{t,j} = 1 - 2\alpha_j^{(t+1)},$$

which is equivalent to what we needed to show. $\square$

The following lemma will be used throughout. It shows that for any interval $[j, j+1]$, once there is some $x_i^{(t)} \in [j, j+1]$, the values of $M_{t,j}$ and $\alpha_j^{(t+1)}$ will remain the same in future iterations, as long as the MLE takes the form $h_{\boldsymbol{\alpha}}$.

**Lemma F.3.** *Under Construction 2, for any $j \in \mathbb{N} \cup \{0\}$ if there exists some $t_j \in \mathbb{N} \cup \{0\}$ with $M_{t_j,j} > 0$, then $\forall t > t_j$, if $s^{(t_j+1)}, \dots, s^{(t)} = 0$,*

1. $M_{t,j} = M_{t_j,j}$,

2. $\alpha_j^{(t+1)} = \alpha_j^{(t_j+1)} = \frac{1}{2}(1 - M_{t,j})$.

*Proof.* We prove the claim by induction on $t$. The case of $t = t_j$ is trivial.

Now, assume the claim holds for some time $t - 1$. Then $\alpha_j^{(t)} = \alpha_j^{(t_j+1)}$, so following Remark 1, all new samples $X^{(t)}$ that are inside the interval $[j, j+1]$ must also be inside the interval

$$
\left[ j, j + 1 - 2\alpha_j^{(t)} \right] = \left[ j, j + 1 - 2\alpha_j^{(t_j+1)} \right]
$$
$$
= \left[ j, j + M_{t_j,j} \right],
$$

where the last equality follows from Lemma F.2. Hence, no new sample in $[j, j+1]$ can exceed $j + M_{t_j,j}$, which by the induction hypothesis was already the maximum. Thus

$$
M_{t,j} = M_{t-1,j} = M_{t_j,j},
$$

and applying Lemma F.2 again gives

$$
\alpha_j^{(t+1)} = \frac{1 - M_{t,j}}{2} = \alpha_j^{(t_j+1)}.
$$

This completes the induction. $\qquad\square$

**Lemma F.4.** *Under Construction 2, let $j, t \in \mathbb{N}$ and $u := (1 - \alpha_j^{(t)}) \prod_{k=0}^{j-1} \alpha_k^{(t)} > 0$. For any $\delta \in (0, 1)$ let*

$$
q := 2 + e^2 u n + 2 \log \left( \frac{1}{\delta} \right),
$$

*let $B$ denote the event that $\exists i \in [n]$ s.t $x_i^{(t)} \in [j, j+1]$ and for any $q \in \mathbb{N}$, let $A_q$ denote the event that $\left| \{ i \in [n] \mid x_i^{(t)} \in [j, j+1] \} \right| \geq q$. Then if $s^{(t)} = 0$,*

$$
\mathbb{P} \left( A_q \mid \theta^{(1)}, \dots, \theta^{(t)}, B \right) \leq \delta.
$$

*Proof.* By construction, for any $i \in [n]$, if $s^{(t)} = 0$ (so that $p_{\theta^{(t)}}$ is of the form $h_{\boldsymbol{\alpha}^{(t)}}$) it holds that

$$
\mathbb{P} \left( x_i^{(t)} \in [j, j+1] \mid \theta^{(1)}, \dots, \theta^{(t)} \right) = (1 - \alpha_j^{(t)}) \prod_{k=0}^{j-1} \alpha_k^{(t)} = u > 0,
$$

Let $b_i$ be 1 if $x_i^{(t)} \in [j, j+1]$ and 0 otherwise. Then conditioned on $\theta^{(1)}, \dots, \theta^{(t)}$, $b_i$ are i.i.d. Bernoulli random variables with parameter $u$, so applying Lemma F.7 completes the proof. $\qquad\square$

**Lemma F.5.** *Under Construction 2, let $j \in \mathbb{N} \cup \{0\}$ and suppose that there exists some $t_j$ such that $M_{t_j,0}, \dots, M_{t_j,j} > 0$. Let $u := (1 - \alpha_{j+1}^{(t_j+1)}) \prod_{k=0}^{j} \alpha_k^{(t_j+1)}$. For any $\delta \in (0, 1)$, letting*

$$
t_{j+1} := \left\lceil t_j + 1 + \frac{2 \log \left( \frac{2}{\delta} \right)}{n \prod_{k=0}^{j} \alpha_k^{(t_j+1)}} \right\rceil,
$$

*and*

$$
q := \left\lceil 2 + e^2 u n + 2 \log \left( \frac{4}{\delta} \right) \right\rceil,
$$

*then it holds with probability at least $1 - \delta$ that either $s^{(t)} = 1$ for some $t \in \{t_j + 1, \dots, t_{j+1}\}$ or*

$$
0 < M_{t_{j+1},j+1} \leq 1 - \frac{\delta}{4q}.
$$

*Proof.* For any $k \in \{0, \ldots, j\}$, by the assumptions that $M_{t_j,k} > 0$, Lemma F.3 states that for all $t \geq t_j$, if $s^{(t_j+1)}, \ldots, s^{(t+1)} = 0$ then $\alpha_k^{(t+1)} = \alpha_k^{(t_j+1)} = \frac{1}{2}(1 - M_{t_j,k}) > 0$.

For any $t, i$ let $b_{t,i}$ be the bernoulli random variables that take the value of 1 if $x_i^{(t)} \in [j+1, j+2]$ and 0 else. By Remark 1, $b_{t,i}$ are $\mathrm{Ber}(u)$ random variables. Let $A_t$ denote the event that $\forall b_{t,i} = 0$. For any $t \geq t_j + 1$, $x_i^{(t)}$ are i.i.d. when conditioned on $\theta^{(1)}, \ldots, \theta^{(t)}$, and when $s^{(t)} = 0$, we get

$$\mathbb{P}\left(A_t \mid \theta^{(1)}, \ldots, \theta^{(t)}\right) = (1-u)^n \leq \exp\left(-nu\right),$$

Notice in particular that $A_t$ depends only on $\theta^{(1)}, \ldots, \theta^{(t_j+1)}$ and $s^{(t_j+1)}, \ldots, s^{(t)}$. So applying this argument inductively for each $t \in \{t_j + 1, \ldots, t_{j+1}\}$ we get that

$$\mathbb{P}\left(\exists t \in \{t_j + 1, \ldots, t_{j+1}\} \text{ s.t } M_{t,j+1} > 0\right) \geq 1 - \exp\left(-(t_{j+1} - t_j - 1)nu\right) \geq 1 - \frac{\delta}{2},$$

where the last inequality follows from the choice of $t_{j+1}$ and the fact that $\alpha_{j+1}^{(t_j+1)} \leq 1/4$.

By Lemma F.3, if $M_{t,j+1} > 0$ for some $t \in \{t_j + 1, \ldots, t_{j+1}\}$ and if $s^{(t)}, \ldots, s^{(t_{j+1})} = 0$ then $M_{t_{j+1},j+1} > 0$. In summary, we have given the lower bound on $M_{t_{j+1},j+1}$ needed for the lemma with probability at least $1 - \delta/2$.

We now move on to the upper bound of the lemma. Suppose that there exists a $\tau$ which is the first timestep for which $M_{\tau,j+1} > 0$ or $s^{(\tau)} = 1$. If $s^{(\tau)} = 1$ we are done, so assume it is 0. Let $B$ denote the event that $\exists i \in [n]$ s.t $x_i^{(\tau)} \in [j+1, j+2]$ and let $A$ denote the event that $\left|\{i \in [n] \mid x_i^{(\tau)} \in [j+1, j+2]\}\right| \leq q$. We want to bound $M_{\tau,j+1}$, where we must condition on the fact there is at least one sample at time $\tau$ that reached interval $j$. Recall that by Lemma F.3, $\alpha_k^{(\tau)} = \alpha_k^{(t_j+1)}$ for all $k \leq j$. By Lemma F.4, using our choice of $q$ we obtain

$$\mathbb{P}\left(A \mid \theta^{(t_1)}, \ldots, \theta^{(\tau)}, B\right) \geq 1 - \frac{\delta}{4}.$$

Now suppose that this event indeed holds, so there are at most $q$ samples that land inside the interval $[j+1, j+2]$ at time $\tau$. Since $x_i^{(\tau)}$ are i.i.d. (when conditioned on $\theta^{(1)}, \ldots, \theta^{(\tau)}$) those that land in interval $[j+1, j+2]$ are distributed within the interval as i.i.d. uniform random variables on $\left[0, 1 - 2\alpha_{j+1}^{(\tau)}\right]$ (which is included in $[0,1]$), so letting $z_1 \ldots, z_q$ be i.i.d. uniform random variables on $[0,1]$, by Lemma F.8 it holds that

$$\mathbb{P}\left(M_{\tau,j+1} \leq 1 - \frac{\delta}{4q}\right) \geq \mathbb{P}\left(\max_{i \in [q]} z_i \leq 1 - \frac{\delta}{4q}\right) \geq 1 - \frac{\delta}{4}.$$

So overall, the desired bounds hold with probability at least $1 - \delta$. $\qquad\square$

**Lemma F.6.** *Under Construction 2, for any $J \in \mathbb{N}$ and for any $\delta \in (0,1)$, let*

$$t_J := \left(\frac{C \log\left(\frac{4}{\delta}\right)}{\delta}\right)^J,$$

*where $C > 0$ is some absolute constant. Then with probability at least $1 - \delta$, there exists some $t \leq t_J$ for which at least one of the following holds:*

1. *$s^{(t)} = 1$.*

2. *$M_{t_J,J} > 0$.*

*Proof.* We begin by analyzing $\theta^{(1)}$. By Lemma F.2,

$$\alpha_0^{(1)} = \frac{1 - M_{0,0}}{2}.$$

By construction, $p_{\theta^{(0)}}$ is the uniform distribution on $[0, 1]$, so $M_{0,0}$ is the maximum of $n$ i.i.d. standard uniform random variables. We thus use Lemma F.8 to bound $M_{0,0}$; so it holds with probability at least $1 - \delta/2$ that

$$1 - \frac{\log\left(\frac{2}{\delta}\right)}{n} \leq M_{0,0} \leq 1 - \frac{\delta}{2n}, \qquad \frac{\delta}{4n} \leq \alpha_0^{(1)} \leq \frac{\log\left(\frac{2}{\delta}\right)}{2n}.$$

By Lemma F.3, this also means that for every $t \geq 0$,

$$1 - \frac{\log\left(\frac{2}{\delta}\right)}{n} \leq M_{t,0} \leq 1 - \frac{\delta}{2n}, \qquad \frac{\delta}{4n} \leq \alpha_0^{(t+1)} \leq \frac{\log\left(\frac{2}{\delta}\right)}{2n}. \tag{30}$$

If $s^{(1)} = 1$ we are done. Assume not. We now move on to bounding $\alpha_j$ for $j > 0$. Set $t_0 := 0$, and for every $j \in [J]$ we define

$$t_j := \left\lceil t_{j-1} + 1 + \frac{2 \log\left(\frac{4J}{\delta}\right)}{n \prod_{k=0}^{j-1} \alpha_k^{(t_{j-1}+1)}} \right\rceil, \tag{31}$$

and

$$q_j := 3 + 2e^2 \prod_{k=0}^{j-1} \alpha_k^{(t_j+1)} n + \log\left(\frac{4}{\delta}\right).$$

Note that the $q_j$ defined here is slightly larger than the one defined in Lemma F.5 as $(1 - \alpha_{j+1}^{(t_j+1)}) < 1$. By Lemma F.5 and Lemma F.3 for any $j \in [J]$, if $M_{t_{j-1},0}, \ldots, M_{t_{j-1},j-1} > 0$, with probability at least $1 - \delta/(2J)$, either there exists some $t \leq t_j$ with $s^{(t)} = 1$ or

$$0 < M_{t_j,j} < 1 - \frac{\delta}{4q_j}. \tag{32}$$

Note that by Lemma F.3, the same bound holds for any $t \geq t_j$ such that $s^{(t_j)}, \ldots, s^{(t)} = 0$. It was already shown for $j = 0$ that $M_{t_0,0} > 0$, so for each $j \in [J]$, conditioning on $t_0, \ldots, t_{j-1}$ it holds with probability at least $1 - j\delta/(2J)$ that either there is some $t \leq t_j$ with $s^{(t)} = 1$, or the bound on $M_{t_j,j}$ given in Eq. (32) holds. Applying the union bound, this is true for all $j \in [J]$ with probability at least $1 - \delta/2$. From now suppose that for all $t \leq t_J$, $s^{(t)} = 0$ (otherwise we are done).

We now move to bounding the $q_j$ terms. Using the bounds on $\alpha_0^{(t_j+1)}$ from Eq. (30), we have

$$q_j \leq 3 + \frac{e^2 \log\left(\frac{2}{\delta}\right)}{2} \prod_{k=1}^{j-1} \alpha_k^{(t_j+1)} + \log\left(\frac{4}{\delta}\right)$$

$$\leq 3 + \frac{e^2 \log\left(\frac{2}{\delta}\right)}{2} + \log\left(\frac{4}{\delta}\right) \leq C'\left(1 + \log\left(\frac{4}{\delta}\right)\right),$$

for some suitable constant $C' > 0$, where we used that $\alpha_k^{(t_j+1)} \in [0, 1/4]$.

As such, by Lemma F.2 and Lemma F.3, it holds for all $j \in [J]$ that

$$\alpha_j^{(t_j+1)} = \frac{1}{2}\left(1 - M_{t_j,j}\right) \geq \frac{\delta}{8q_j} \geq \frac{\delta}{8C'\left(1 + \log\left(\frac{4}{\delta}\right)\right)}.$$

So using this bound, Eq. (30), and taking $C = \max(8C', 1)$, for any $j \in [J]$, the product can be bounded as

$$n \prod_{k=0}^{j-1} \alpha_k^{(t_j+1)} \geq \frac{\delta}{4} \cdot \left(\frac{\delta}{C\left(1 + \log\left(\frac{4}{\delta}\right)\right)}\right)^{j-1} \leq \left(\frac{\delta}{C\left(1 + \log\left(\frac{4}{\delta}\right)\right)}\right)^j.$$

Overall, Eq. (31) leads to

$$t_J \leq 2J + 2\log\left(\frac{4J}{\delta}\right)\sum_{j=1}^{J}\frac{1}{n\prod_{k=0}^{j-1}\alpha_k^{(t_j+1)}}$$

$$\leq 2J + \log\left(\frac{4J}{\delta}\right)\sum_{j=1}^{J}\left(\frac{C\left(1 + \log\left(\frac{4}{\delta}\right)\right)}{\delta}\right)^j.$$

The right-hand side is a geometric series of the form $\sum_{j=1}^{J} r^j$ for $r > 2$. Furthermore, for any $r \geq 2$ a geometric series satisfies $\sum_{j=1}^{J} r^j \leq 2r^J$. Using this, we obtain

$$t_J \leq 2J + 4\log\left(\frac{4J}{\delta}\right)\left(\frac{C\left(1 + \log\left(\frac{4}{\delta}\right)\right)}{\delta}\right)^J.$$

Replacing $C$ by a suitable larger constant $C$, this can be upper bounded as

$$t_J \leq \left(\frac{C\log\left(\frac{4}{\delta}\right)}{\delta}\right)^J.$$

$\square$

**Theorem 5.2.** *Let $\phi : (0, \infty) \to (0, \infty)$ be any strictly monotonically increasing function such that $\lim_{n\to\infty} \phi(n) = \infty$. Then there exists an absolute constant $C > 0$, a set $\Theta$, $\theta^\star \in \Theta$ and a TV-consistent family of distributions $P_\Theta$ (which depends on $\phi$), such that for any $\delta \in (0, 1)$, $n \in \mathbb{N}$, it holds with probability at least $1 - \delta$ that*

$$\mathrm{TV}\left(p_{\theta^{(T)}}, p_{\theta^\star}\right) \geq \frac{3}{8} \quad \textit{for some} \quad T \leq \left\lceil\frac{C}{\delta}\log\left(\frac{4}{\delta}\right)\cdot\max\left(\phi(n), 1\right)\right\rceil.$$

*Proof.* TV-consistency of $P_\Theta$ and existence of $\theta^{(t)}$ for any $t \in \mathbb{N}$ are given by Lemma F.1.

For any $J > 1$, and $x \in [0, 1]$, $g_{\beta,J}(x) = \frac{1}{2J} \leq \frac{1}{4}$ but $p_{\theta^{(0)}}(x) = 1$. So the TV distance between any $g_{\beta,J}(x)$ and $p_{\theta^{(0)}}(x)$ is lower bounded as

$$\frac{1}{2}\int_{\mathbb{R}}|g_{\beta,J}(x) - p_{\theta^{(0)}}(x)|\,dx \geq \frac{1}{2}\int_0^1\left|\frac{1}{4} - 1\right|dx \geq \frac{3}{8}.$$

As such, it suffices to show that there exists some time $T$ such that with the desired probability, $g_{\beta,J}$ is chosen as the MLE.

As mentioned in Remark 1, if there is some non-negative integer $j(x)$ such that $x \in [j(x), j(x) + 1 - 2\alpha_{j(x)}]$, then

$$\log\left(h_{\boldsymbol{\alpha}}(x)\right) = \log\left(1 + \frac{\alpha_{j(x)}}{1 - 2\alpha_{j(x)}}\right) + \sum_{k=0}^{j(x)-1}\log\left(\alpha_k\right). \tag{33}$$

If no such $j(x)$ exists, then $\log\left(h_{\boldsymbol{\alpha}}(x)\right)$ is undefined. Note that the first term is increasing in $\alpha_{j(x)}$ and the second is negative (because $\alpha_k \leq 1/4$). As such,

$$\sum_{t=0}^{T}\sum_{i=1}^{n}\log\left(h_{\boldsymbol{\alpha}}(x_i^{(t)})\right) \leq nT\log\left(1 + \frac{\frac{1}{4}}{\frac{1}{2}}\right) < nT. \tag{34}$$

For the PDFs $g_{\beta,J}$, we have

$$\log\left(g_{\beta,J}(x)\right) = \begin{cases} -\log(2J) & x \in [0, J]\setminus[J - \beta, J - \beta + f(J)] \\ -\log(2J) + \log\left(\frac{1}{2f(J)}\right) & x \in [J - \beta, J - \beta + f(J)] \\ \text{undefined} & \text{else} \end{cases}.$$

We now show that if $T$ and $J$ are such that $T$ is sufficiently large and there is some sample in the interval $[J-1, J]$, then a function of the form $g_{\beta, J}$ will be the MLE. We will show that there exist some $g_{\beta, J}$ for which the log likelihood is bigger than for all PDFs of the form $h_{\alpha}$. The existence of the MLE implies that there must be some function of the form $g_{\beta, J}$ that is the MLE.

Now fix any $J$ which will be specified later, and suppose momentarily that for some $T \in \mathbb{N}$, $M_{T, J-1} > 0$ (meaning that there exists some sample in $[J-1, J]$).

Let $\beta_J := 1 - M_{T, J-1} + \frac{1}{2} f(J)$ such that $J - \beta_J = J - 1 + M_{T, J-1} - \frac{1}{2} f(J)$ and as such, by the definition of $M_{T, J-1}$ there must be some point in $[J - \beta_J, J - \beta_J + f(J)]$. Note that for any $J \in \mathbb{N}$, since $f$ is assumed to satisfy $f(J) \leq \frac{1}{2}$ it holds that $\log\left(\frac{1}{2f(J)}\right) \geq 0$, and thus

$$\sum_{t=0}^{T} \sum_{i=1}^{n} \log\left(g_{\beta_J, J}(x_i^{(t)})\right) \geq -Tn\log(2J) + \log\left(\frac{1}{2f(J)}\right). \tag{35}$$

In particular, to ensure the log likelihood of $g_{\beta_J, J}$ is bigger than for any $h_{\alpha}$, it suffices for the right-hand side of Eq. (35) upper bound the right-hand side of Eq. (34). So we want:

$$\log\left(\frac{1}{2f(J)}\right) \geq nT\left(1 + \log(2J)\right) = nT\log\left(2eJ\right).$$

Taking the exponential of both sides and rearranging, the above is equivalent to

$$f(J) \leq \frac{1}{2(2eJ)^{nT}}. \tag{36}$$

To that end, by Lemma F.6, for some absolute constant $C > 0$, letting

$$T := \psi(J), \qquad \forall a \in \mathbb{R}, \ \ \psi(a) := \left(\frac{C\log\left(\frac{4}{\delta}\right)}{\delta}\right)^{a-1},$$

it holds with probability at least $1 - \delta$, that either $s^{(t)} = 1$ for some $t \leq T$ or $M_{T, J-1} > 0$. If the first holds, we are done, so assume the latter.

Now, for any strictly monotonically increasing function $\phi : (0, \infty) \to (0, \infty)$ with $\lim_{n \to \infty} \phi(n) = \infty$, let

$$f(J) := \frac{1}{2(2eJ)^{\phi^{-1}(\psi(J))\psi(J)}}. \tag{37}$$

Then to ensure Eq. (36) is satisfied, we need $\phi^{-1}(\psi(J)) \geq n$, or equivalently, $J \geq \max\left(\psi^{-1}(\phi(n)), 2\right)$ (where the 2 is because our domain includes only $J \geq 2$). In particular, we take $J := \max\left(\lceil \psi^{-1}(\phi(n)) \rceil, 2\right)$. If $J = 2$, $T = \psi(2) = \frac{C\log\left(\frac{4}{\delta}\right)}{\delta}$, and otherwise we can bound $T$ as

$$T = \psi(J) = \psi\left(\lceil \psi^{-1}(\phi(n)) \rceil\right) \leq \psi(\psi^{-1}(\phi(n)) + 1)$$
$$= \left(\frac{C\log\left(\frac{4}{\delta}\right)}{\delta}\right) \cdot \psi\left(\psi^{-1}(\phi(n))\right) = \left(\frac{C\log\left(\frac{4}{\delta}\right)}{\delta}\right)\phi(n).$$

In summary, we have shown that at some timestep up to $T$, it holds with probability at least $1 - \delta$ that the TV distance is at least $3/8$. $\qquad \square$

## F.1 Auxiliary Lemmas

**Lemma F.7.** *For all $i \in [n]$ let $b_i \sim Ber(u)$ be i.i.d. Bernoulli random variables with parameter $u$. Then for any $\delta \in (0, 1)$,*

$$\mathbb{P}\left(\sum_{i=1}^{n} b_i \geq 2 + e^2 un + 2\log\left(\frac{1}{\delta}\right) \mid \sum_{i=1}^{n} b_i \geq 1\right) \leq \delta.$$

*Proof.* Since $b_i$ are i.i.d., using the inequality $1 - x \leq \exp(-x)$ for any $x$,

$$\mathbb{P}\left(\sum_{i=1}^n b_i \geq 1\right) = 1 - (1-u)^n \geq 1 - \exp(-un).$$

By Chernoff's inequality (c.f. [Vershynin, 2018]) and the chain rule of probability for any $q > un$,

$$\mathbb{P}\left(\sum_{i=1}^n b_i \geq q \mid \sum_{i=1}^n b_i \geq 1\right) = \frac{\mathbb{P}\left(\sum_{i=1}^n b_i \geq q\right)}{\mathbb{P}\left(\sum_{i=1}^n b_i \geq 1\right)} \leq \frac{\exp(-un)}{1-\exp(-un)}\left(\frac{eun}{q}\right)^q \leq \frac{1}{un}\left(\frac{eun}{q}\right)^q,$$

(38)

where the last inequality uses that $\frac{e^{-x}}{1-e^{-x}} \leq 1/x$ for any $x > 0$. Now we split into two cases depending on $un$. First, if $un \leq \frac{4}{e^2}\delta < 1$, for any $q \geq 2$, Eq. (38) becomes

$$\mathbb{P}\left(\sum_{i=1}^n b_i \geq q \mid \sum_{i=1}^n b_i \geq 1\right) \leq \frac{1}{un}\left(\frac{eun}{q}\right)^q = \left(\frac{e}{q}\right)^q (un)^{q-1} \leq \frac{e^2}{4}un \leq \delta.$$

On the other hand, if $un > \frac{4}{e^2}\delta$, taking $q \geq 2 + e^2 un + 2\log\left(\frac{1}{\delta}\right)$ (which in particular ensures that $\left(\frac{eun}{q}\right) \leq 1/e$ and $q \geq 2 + 2\log\left(\frac{1}{\delta}\right)$), Eq. (38) becomes

$$\mathbb{P}\left(\sum_{i=1}^n b_i \geq q \mid \sum_{i=1}^n b_i \geq 1\right) \leq \frac{1}{un}\exp\left(-2 - 2\log\left(\frac{1}{\delta}\right)\right) < \frac{e^2}{4\delta}\frac{1}{e^2}\delta^2 < \delta.$$

In either case, $q \geq 2 + e^2 un + 2\log\left(\frac{1}{\delta}\right)$ suffices to ensure that desired bound. $\qquad\square$

**Lemma F.8.** *For $n \in \mathbb{N}$ let $x_1, \ldots, x_n \sim U([0,1])$ be i.i.d. uniform $[0,1]$ random variables.*

    *1. For any $\delta > 0$, it holds with probability at least $1 - \delta$ that*

$$\max_{i \in [n]} x_i \leq 1 - \frac{\delta}{n}.$$

(39)

    *2. For any $\delta > 0$, it holds with probability at least $1 - \delta$ that*

$$\max_{i \in [n]} x_i \geq 1 - \frac{\log\left(\frac{1}{\delta}\right)}{n}.$$

(40)

*As a result, for any $\delta > 0$, it holds with probability at least $1 - \delta$ that*

$$1 - \frac{\log\left(\frac{2}{\delta}\right)}{n} \leq \max_{i \in [n]} x_i \leq 1 - \frac{\delta}{2n}.$$

(41)

*Proof.* Since $x_i$ are i.i.d., the CDF of $\max_{i \in [n]} x_i$ is given by

$$\mathbb{P}\left(\max_{i \in [n]} x_i \leq 1 - u\right) = \prod_{i=1}^n \mathbb{P}\left(x_i \leq 1 - u\right) = (1-u)^n.$$

To prove Eq. (39), by Bernoulli's inequality $(1-u)^n \geq 1 - un$, so it suffices to take $u = \frac{\delta}{n}$.

To prove Eq. (40), we use the well known inequality $1 - u \leq \exp(-u)$ to obtain

$$\mathbb{P}\left(\max_{i \in [n]} x_i \geq 1 - u\right) = 1 - (1-u)^n \geq 1 - \exp(-un).$$

Taking $u = \frac{1}{n}\log\left(\frac{1}{\delta}\right)$ completes the proof.

Eq. (41) follows from Eq. (40) and Eq. (39) with $\delta/2$ and the union bound. $\qquad\square$

The following lemma gives a version of the uniform law of large numbers that is suited for TV-consistency. We note that the conditions can be made even milder (c.f. [Tauchen, 1985]), and are relatively similar to those of [Wald, 1949, Redner, 1981].

**Lemma F.9** (Newey and McFadden [1994] Lemma 2.4). *Let $\Theta \subseteq \mathbb{R}^d$ be compact, $\theta^{(0)} \in \Theta$, let $\{x_i\}_{i=1}^n \sim p_{\theta^{(0)}}$ be i.i.d. and let $f(x, \theta)$ be a function which for any $\theta \in \Theta$ is measurable, continuous for almost all $x$s, and satisfies $|f(x, \theta)| \leq \phi(x)$ for some function $\phi(x)$ with $\mathbb{E}_{x \sim p_{\theta^{(0)}}}[\phi(x)] < \infty$. Then $\mathbb{E}[f(x, \theta)]$ is continuous in $\theta$ and*

$$\sup_{\theta \in \Theta} \left| \frac{1}{n} \sum_{i=1}^n f(x_i, \theta) - \mathbb{E}_{x \sim p_{\theta^{(0)}}}[f(x, \theta)] \right| \xrightarrow[n \to \infty]{\mathbb{P}} 0.$$

**Lemma F.10.** *Let $\Theta \subseteq \mathbb{R}^d$ be compact, $\bar{\theta} \in \Theta$, and assume that for any $\theta \in \Theta$, $\log(p_\theta(\mathbf{x}))$ is measurable, continuous for almost all $\mathbf{x}$, and satisfies $|\log(p_\theta(\mathbf{x}))| \leq \phi(\mathbf{x})$ for some function $\phi(\mathbf{x})$ with $\mathbb{E}_{\mathbf{x} \sim p_{\bar{\theta}}}[\phi(\mathbf{x})] < \infty$. Then if for any $n$, there exists an MLE $\hat{\theta}^{(n)}$ with respect to $n$ i.i.d. samples from $\bar{\theta}$, it holds that*

$$\mathrm{TV}\left(p_{\bar{\theta}}, p_{\hat{\theta}^{(n)}}\right) \xrightarrow[n \to \infty]{\mathbb{P}} 0.$$

*Proof.* By Lemma F.9, for any $\delta, \epsilon > 0$, there is some $n_0 \in \mathbb{N}$ such that for all $n \geq n_0$,

$$\sup_{\theta \in \Theta} \mathbb{P}\left( \left| \ell(\theta) - \mathbb{E}_{\mathbf{x} \sim p_{\bar{\theta}}}[-\log(p_\theta(\mathbf{x}))] \right| \leq \epsilon^2 \right) \geq 1 - \frac{\delta}{2}.$$

$\hat{\theta}^{(n)}$ minimizes $\ell$, implying $\ell(\hat{\theta}^{(n)}) \leq \ell(\bar{\theta})$ and thus with probability at least $1 - \delta$,

$$0 \geq \ell(\hat{\theta}^{(n)}) - \ell(\bar{\theta}) \geq \mathbb{E}_{\mathbf{x} \sim p_{\bar{\theta}}}[-\log(p_{\hat{\theta}^{(n)}}(\mathbf{x})) + \log(p_{\bar{\theta}}(\mathbf{x}))] - 2\epsilon^2 = D_{\mathrm{KL}}\left(p_{\bar{\theta}} \,\|\, p_{\hat{\theta}^{(n)}}\right) - 2\epsilon^2.$$

Rearranging and using Pinsker's inequality,

$$\mathrm{TV}\left(p_{\bar{\theta}}, p_{\hat{\theta}^{(n)}}\right) \leq \sqrt{\frac{1}{2} D_{\mathrm{KL}}\left(p_{\bar{\theta}} \,\|\, p_{\hat{\theta}^{(n)}}\right)} \leq \sqrt{\frac{1}{2} 2\epsilon^2} = \epsilon.$$

$\square$

## G   Experiments

Figure 1: MLE for a one-dimensional Gaussian distribution.

There are by now many experiments in the literature that support our finding from Theorem 4.1 [Alemohammad et al., 2024, Gerstgrasser et al., 2025, Dey and Donoho, 2024]. In particular, in those papers, the error does not increase much from iteration to iteration when synthetic data is added gradually.

Rather than repeating experiments, we analyze the difference between exact MLE solutions and those that are obtained via optimization. To this end, we pick several families whose MLE has a known closed form. These include a Gaussian (where the parameters are the mean and std) in Fig. 1, Exponential distribution in Fig. 2, and a family of Beta distributions with PDFs given by

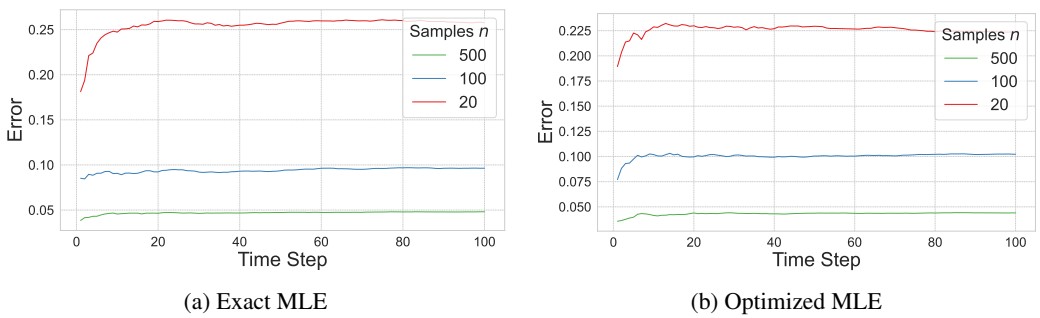

(a) Exact MLE | (b) Optimized MLE

Figure 2: MLE for a one-dimensional Exponential distribution.

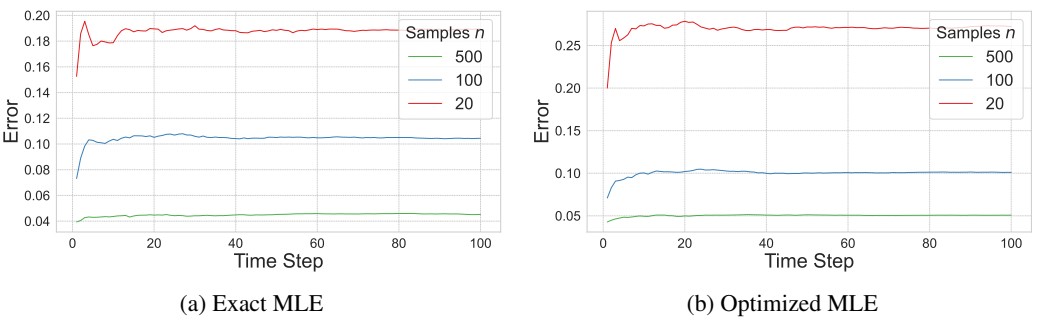

(a) Exact MLE | (b) Optimized MLE

Figure 3: MLE with respect to a Beta distribution family with PDFs given by $p(x; \theta) = \theta x^{\theta-1}$ for $\theta > 0$ and $x \in (0, 1)$.

$p(x; \theta) = \theta x^{\theta-1}$ for $\theta > 0$ and $x \in (0, 1)$ in Fig. 3. The real parameters are $\theta_0 = (\mu = 0, \sigma = 1)$ for the Gaussian and $\theta_0 = 1$ for the other distributions. When optimizing numerically for the MLE, we use scipy.optimize.minimize on the negative log likelihood to find the parameters. We opt for this built-in function to remove any uncertainty regarding the quality of the optimization code itself. We take the number of samples to be one of $20, 50$, or $100$. We run the iterative MLE algorithm as specified in the paper for up to $T = 100$. All values are averaged over 50 runs. The error is measured by the norm relative to the real parameters, meaning $\left\| \theta^{(t)} - \theta_0 \right\|$. In all cases, the error at all timesteps is similar to the error at time 1, as our theory would suggest. Furthermore, we observe the model (non)-collapse behavior between the exact MLE and the optimized one to be similar.

We also consider various $\theta_0$ going from $0.1$ to $1$ for the Beta distribution, where a smaller $\theta_0$ corresponds to a neighborhood of the parameters that are less smooth. In all cases, we plot the ratio between the error at time $T$ to the error at time 1. For $\theta = 1$, the error increases by a factor of only $1.25$ across 100 iterations, but for the "less smooth" $\theta_0 = 0.1$, the error increases by a factor of $3.27$. These confirm that our negative results hint at a more general phenomenon and support our results.

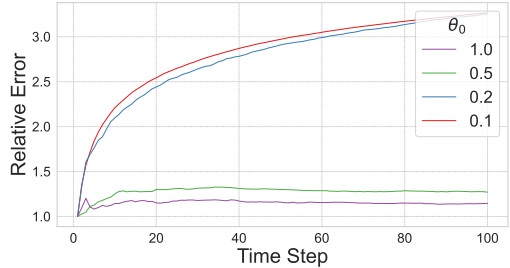

Figure 4: MLE with respect to a Beta distribution family with PDFs given by $p(x; \theta) = \theta x^{\theta-1}$ for $\theta > 0$ and $x \in (0, 1)$, for various choices of real parameter $\theta_0$.

