# OpenReview forum: "When Models Don’t Collapse: On the Consistency of Iterative MLE"
_NeurIPS.cc/2025/Conference — NeurIPS 2025 poster_

### Official Review · Reviewer_15P5 · 2025-06-22

**Clarity:** 4
**Significance:** 3
**Originality:** 3
**Rating:** 4
**Confidence:** 4

**Summary:**

- Theoretically studies model collapse for maximum likelihood estimation (MLE), focuses on setting where synthetic data is added to the original dataset
- Under standard assumption, establish that model collapse can be avoided even as the fraction of real data vanishes

**Questions:**

- Introduction lines 42: Regarding Dey and Donoho: “Because in their setting, there is always a constant fraction of training data that is real.” Looking at Workflow 1 of Dey and Donoho, I do not believe this is true but I am possibly misunderstanding. Can the authors please clarify?

**Ethical Concerns:**

["NO or VERY MINOR ethics concerns only"]

**Final Justification:**

I thought the paper was good overall, and the authors have agreed to make a few minor adjustments (primarily towards the discussion and context) that I think will improve the paper.

I still feel like the paper is on the shorter side. While its contents are high quality, they feel stretched out and the manuscript overall doesn't feel like a full conference paper.

**Limitations:**

Yes. I think the authors could more explicitly state that the contributions are entirely theoretical and more work is necessary to connect to empirical phenomena.

**Quality:**

3

**Strengths And Weaknesses:**

- Strength: The paper is extremely well written: easy to follow, easy to understand
- Strength: Theorem 4.1 is clean, elegant, meaningful
- Weakness: Section 5 is very well motivated but the results are a bit odd. The distributions that appear feel carefully constructed and “unnatural”. It’s difficult to know what to make of the significance of Theorem 5.1 and Theorem 5.2 because I don’t know how general these distributions are (or what the salient properties are). In comparison, Theorem 4.1 feels much more general. One would like to know: are large scale deep generative models likely to produce model collapse? Does web-scale data / synthetic data have such distributional properties that would cause model collapse? I don't know after reading Section 5.
- Weakness: The paper feels overall short. Section 3 is a lot of notation, preliminaries, algorithm 1, assumption 1, assumption 2, assumption 3. Most of Section 4 is a sketch of the proof (I like hearing the sketch, but I could imagine some may prefer it to be deferred to the appendix). Section 5 similarly spends a lot of time on sketching proofs.
- (Semi)Weakness: Many papers concerning model collapse reach (softly) contradictory conclusions. For example, Strong Model Collapse argues "Our results show that even the smallest fraction of synthetic data (e.g., as little as 1 per 1000) can still lead to model collapse: larger and larger training sets do not enhance performance." I think this paper could be improved by connecting more to prior work and explaining whether this paper's contributions and other papers' contributions are consistent and/or inconsistent, and explaining why different conclusions are reached.

---

> ### Author Rebuttal · Authors · 2025-07-29
>
> We thank the reviewer for their constructive comments and their time. Below, we address all of the comments raised by the reviewer. If there are any concerns remaining after this response, please let us know so that we can address them.
>
> - "are large scale deep generative models likely to produce model collapse?": Generally speaking, neural networks do fit the setting of Theorem 4.1 (the positive result). For the regularity conditions, a constant support (Assumption 1.C) is trivially satisfied with standard architectures, since the softmax function, which is used to assign probabilities, is always non-zero. The fact that the models are unique (Assumption 1.B) is made for convenience, and is not essential, as discussed in the paper. Positive definiteness of the Fisher information matrix can also be shown to be satisfied in various settings, and there are entire papers devoted to analyzing the eigenvalues of this matrix (e.g. [1]). Smoothness assumptions can be satisfied in various settings, especially when using techniques such as weight decay. Of course, the exact bounds would depend on the architecture and the setting.
>
> - Regarding section 5 (the negative results): Even though the constructions are specific, they highlight more general phenomena that are needed for model collapse to occur or to be avoided. In particular, the main difference from Theorem 4.1 is the smoothness assumption: Theorem 5 crucially uses this non-smoothness, and we find it unlikely that a negative example would be possible under the smoothness assumption of Theorem 4.1.
>
> - Connecting to prior work: We agree that many model collapse papers appear to reach differing conclusions, largely caused by the differences in the settings. For example, the excellent “Strong Model Collapse” paper analyzes a particular version of linear regression, which is different from a generative setting as in this paper. Moreover, their setting is non-iterative, and they analyze the test error when the training data contains some samples that are labeled from a linear predictor drawn from a "bad” synthetic distribution that differs from the real one.
> Beyond this paper, another notable difference between past works that can influence model collapse is whether or not synthetic data is *added* to the original dataset, as opposed to *replacing* it. We are happy to expand the discussion in the paper comparing these differences in finer detail.
>
> - Regarding Dey and Donoho: In their paper, the theorems hold in a setting where for each FIXED iteration $t$ (they denote this by $G$ in their paper), the number of samples per iteration $n$ tends to infinity. So the proportion of real data is $1/t$, which is independent of $n$. In contrast, our Theorem 4.1 in particular holds whenever $t$ is sub-exponential in $n$, and the fraction of real data given by $1/t$ may vanish.
>
> We hope our response addresses the reviewer’s comments, and will be happy to further clarify any point if needed.
>
> [1] The Spectrum of the Fisher Information Matrix of a Single-Hidden-Layer Neural Network, Pennington and Worah 2019

---

> ### Comment · Reviewer_15P5 · 2025-08-01
> **Response to Authors**
>
> > "are large scale deep generative models likely to produce model collapse?": Generally speaking, neural networks do fit the setting of Theorem 4.1 (the positive result). For the regularity conditions, a constant support (Assumption 1.C) is trivially satisfied with standard architectures, since the softmax function, which is used to assign probabilities, is always non-zero. The fact that the models are unique (Assumption 1.B) is made for convenience, and is not essential, as discussed in the paper. Positive definiteness of the Fisher information matrix can also be shown to be satisfied in various settings, and there are entire papers devoted to analyzing the eigenvalues of this matrix (e.g. [1]). Smoothness assumptions can be satisfied in various settings, especially when using techniques such as weight decay. Of course, the exact bounds would depend on the architecture and the setting.
>
> I may have missed this in the original manuscript, but please emphasize this point in the Discussion. I like for theoretical models to have connections to real problems, and I think it's good to stay focused on "why do we care about this setting / phenomenon?"
>
> > Regarding section 5 (the negative results): Even though the constructions are specific, they highlight more general phenomena that are needed for model collapse to occur or to be avoided. In particular, the main difference from Theorem 4.1 is the smoothness assumption: Theorem 5 crucially uses this non-smoothness, and we find it unlikely that a negative example would be possible under the smoothness assumption of Theorem 4.1.
>
> Is there intuition for why smoothness matters so much? I would naively think in a (nearly) infinite data limit and for a sufficiently expressive model, non-smoothness of the distribution wouldn't pose a problem.
>
> ## Last Comment
>
> I still feel like the paper is on the shorter side. While its contents are high quality, they feel stretched out and the manuscript overall doesn't feel like a full conference paper.

---

> > ### Author Response · Authors · 2025-08-04
> > **Response to Reviewer 15P5**
> >
> > We thank the reviewer for their comments. We would like to answer your questions and address your remaining concerns.
> >
> > - Connection to real problem: Yes, we are happy to add this to the paper and elaborate on the theoretical importance as we discussed.
> >
> > - Intuition for smoothness: The word “nearly” in your question is important. While access to infinitely many samples would ensure recovering the correct distribution exactly (which would prevent model collapse), for finitely many samples, there will be a small difference between the recovered parameters and the real ones. Our negative theorems show that there are cases where no matter how small this difference is, it is enough for the synthetic data to cause model collapse. Smoothness would prevent such constructions, since, intuitively, similar parameters would lead to the synthetic data being sufficiently similar to real data.
> >
> > - Length of paper: We would like to note that the paper is 40 pages long (33 without the NeurIPS checklist), and this is with an appendix that contains math, and not large figures that take up entire pages. Furthermore, it is common for theoretical papers in NeurIPS and similar conferences to have only one main theorem, whereas this paper has three. We therefore believe that the length of the paper (including the math in the appendix) is **more** than in the average NeurIPS accepted paper. Nevertheless, we will add to the final paper more discussions on the relevance of the setting, the significance of the results, and their consequences. These will increase the amount of non-mathematical content in the main part of the paper, which we believe will address the reviewer’s concern.
> >
> > We hope that we have fully addressed your concern, and are happy to clarify anything if needed.

---

### Official Review · Reviewer_a93t · 2025-06-28

**Clarity:** 3
**Significance:** 2
**Originality:** 3
**Rating:** 4
**Confidence:** 2

**Summary:**

This paper proposes an abstraction for training generative models using synthetic data, formalized in Algorithm 1. Within this framework, the authors conduct a non-asymptotic analysis of distribution estimation under various assumptions. In particular, Section 4 demonstrates that iterative MLE remains consistent under Assumptions 1–3 (i.e., regularity conditions and smoothness), while Section 5 shows that without these assumptions, there exist distribution families for which iterative MLE yields a solution that deviates from the true distribution by a constant gap, indicating inconsistency.

**Questions:**

1. Given the first and second weaknesses mentioned above, could the authors either:

  - propose variants of the iterative MLE algorithm that better reflect practical setups, or

  - justify the current modeling assumptions with concrete real-world examples (e.g., specific use cases where the assumptions approximately hold)?

2. Given the third weakness mentioned above,  I strongly recommend including simulations and, if possible, real-world experiments to illustrate the behavior of iterative MLE.

3. Regarding the analysis of the theoretical results:

  - Are the key distinctions between Theorems 4.1 and 5.1–5.2 primarily the presence or absence of Assumptions 1–3?

  - If so, can the authors clarify why these assumptions are critical to ensuring consistency of iterative MLE (for instance, the smoothness might be technically necessary for controlling cumulative estimation errors across iterations)?

**Ethical Concerns:**

["NO or VERY MINOR ethics concerns only"]

**Final Justification:**

The authors have provided detailed clarifications and supporting materials, which have well addressed my concerns. I adjust my score to 4.

**Limitations:**

The Discussion section currently only addresses the limitation of weakening the assumptions. Given the practical concerns raised above, if the analysis of more realistic training setups is indeed technically intractable or out of scope, I recommend the authors explicitly acknowledge this as a limitation.

**Paper Formatting Concerns:**

None.

**Quality:**

3

**Strengths And Weaknesses:**

**Strengths**

1. The theoretical content and proofs presented in the paper are based on analyses of MLE consistency and normality, which appear sound (though I have not verified the proof details).
2. The logical progression from Section 4 to Section 5 is well-motivated, allowing me to follow the authors’ idea from "classical assumptions yielding positive results" to "relaxed assumptions yielding negative results."
3. The results in Section 5 are particularly interesting, as they highlight concrete scenarios in which iterative MLE fails under more general assumptions (compared to the strong smoothness assumptions in Section 4, which are hard to verify in complex neural networks).

**Weaknesses**

The paper adopts a highly theoretical focus but does not sufficiently connect its analysis to practical generative modeling scenarios (while a purely technical contribution with novel theoretical insights would justify a strong theoretical significance, this paper does not appear to aim for such breakthroughs). Instead, its motivation is to inspire real-world insights. Thus, the practical relevance of the setup and assumptions becomes essential. My main concerns are:

1. Iterative MLE algorithm: The proposed algorithm differs from how synthetic data is used in practice. In real-world settings, synthetic data rarely dominates the training set as much as assumed here (e.g., in foundation models, data is often filtered). Moreover, synthetic data is often deliberately designed rather than directly sampled from model distributions (e.g., instruction-following data, CoT data). These discrepancies reduce the practical relevance of the proposed framework.

2. MLE: Modern large-scale generative models are typically not trained to full MLE convergence (e.g., current LLMs typically stop training while the loss curve is still declining). As such, the assumption of MLE may not reflect actual training dynamics.

3. Experiments: The absence of experiments is a notable limitation. Simulations that validate the theoretical findings under the assumptions in Sections 4 and 5 are necessary, and real-world experiments would significantly enhance the paper's significance.
These concerns are the primary reasons for my current overall score. I would be willing to raise my score if they are adequately addressed.

---

> ### Author Rebuttal · Authors · 2025-07-29
>
> We thank the reviewer for their constructive comments and their time. Below, we address all of the comments raised by the reviewer. If there are any concerns remaining after this response that are preventing the reviewer from raising their score to acceptance, please let us know so that we can address them.
>
> - Theoretical paper: It is important to note that this is, in fact, a theoretical paper and should be judged as such. Of course, the theory aims to better understand a phenomenon that has gained much practical interest in recent years, but it is still a theoretical paper.
>
> - Practical Setting: Thank you for this comment; it is indeed important that the paper accurately conveys the practicality of this setting, as it has indeed become highly relevant recently. We will clarify this in the paper.
> The scenario we are analyzing is synthetic data appearing inadvertently in datasets, due to the rapidly widening use of generative models (this is different than synthetic data that is intentionally designed and added to the training data). The iterative setting that we study (and variations of it) has already gained huge interest with many papers on model collapse over the past few years (e.g. [1-6]. A comprehensive list of papers on the topic is given in [5]), which have even led to news articles in the New York Times and Wall Street Journal.
> As a simple example that demonstrates the importance of such a setting, consider an LLM with some cultural/linguistic biases. People using the LLM would increase the amount of web data with these biases, and it is not easy to filter unless huge chunks of LLM outputs are copy-pasted as-is. Even in such cases, filtering all LLM outputs on large datasets is unrealistic. It is therefore inevitable that these biases will be more present in future training sets and could thus potentially be amplified in future iterations.
>
> - Justification of the current modeling assumptions with concrete examples:
> The modeling assumptions are very general and can capture standard neural network training. We will add to the paper a discussion showing how our theory captures practical settings. In particular, smoothness assumptions can be satisfied as long as the weights of the neural network don't diverge to infinity (indeed, in practice, they typically don't). Positive definiteness of the Fisher information matrix can also be shown to be satisfied in various settings, and there are entire papers devoted to analyzing the eigenvalues of this matrix (e.g. [7]). For the regularity conditions, a constant support (Assumption 1.C) is trivially satisfied in standard architectures, since the softmax function widely used to assign probabilities is always non-zero. The fact that the models are unique (Assumption 1.B) is made for convenience, and is not essential as discussed in the paper.
>
> - Simulations: We will be glad to add simulations to the paper in order to complement our theoretical findings. Note that we cannot directly display them in the rebuttal as per the new NeurIPS policy, but we will add these to the camera-ready version. We also note that there are already some experiments in the literature in our setting (e.g. [2]) which align with our theoretical results, and we will reference these as well.
>
> - Approximate convergence: The extension of the results to approximate stationary points is of course, possible. However, it is common to omit such extensions in generative modeling papers as they unnecessarily over-complicate the analysis, hiding the main points of the theorems. Analysis of the exact MLE is something that has been common for many decades and continues to interest many people in the community. In particular, this is standard in previous papers on model collapse  (e.g. [3], [6]).
>
> - Difference between theorems 4.1 and 5.1-5.2 - Yes, the reason for the difference is the assumptions. Theorem 4.1 follows assumptions that are standard in the MLE literature, and Theorems 5.1-5.2 complement this result by showing what can go wrong when these assumptions are violated.
> The main assumption that is critical is indeed smoothness. There will typically be some difference between the real model and the learned one. If arbitrarily similar parameters can result in models that behave very differently, we believe it is unlikely that a theorem like 4.1 would be possible in general. To prevent uninteresting counter-examples to Theorem 4.1, we require a minimal assumption of consistency in Theorems 5.1-5.2. Finding any cases in which iterative MLE fails in the data-accumulating setting was an open question that interested many people in the model Collapse community, and that we address in our paper.
>
>
>
> We hope our response addresses the reviewer’s comments, and will be happy to further clarify any point if needed.
>
>
> [1] AI models collapse when trained on recursively generated data, Shumailov et al. 2024
>
> [2] Self-Consuming Generative Models Go MAD, Alemohammad et al. 2023
>
> [3] On the stability of iterative retraining of generative models on their own data, Bertrand et al. 2024
>
> [4] Strong Model Collapse, Dohmatob et al. 2025
>
> [5] Position: Model collapse does not mean what you think, Schaeffer et al., 2025
>
> [6] Universality of the π2/6 Pathway in Avoiding Model Collapse, Dey and Donoho, 2024
>
> [7] The Spectrum of the Fisher Information Matrix of a Single-Hidden-Layer Neural Network, Pennington and Worah 2019

---

> > ### Comment · Reviewer_a93t · 2025-08-03
> >
> > Thank you for the response, especially the explanation in *"Justification of the current modeling assumptions with concrete examples"* for the theoretical assumptions listed in the paper. However, my main concern remains. The setting studied in the paper has some key differences from practice. Even if they might be approximations to practice (e.g., the MLE and distribution of synthetic data), they are likely to have a significant impact on the derived results for theoretical work. Considering the above, I will change my score accordingly.

---

> ### Author Response · Authors · 2025-08-06
> **Response to Reviewer a93t (Continued)**
>
> Raw Data # 1: Data that will be used to show convergence, confirmation of Theorem 4.1, and comparisons between exact solutions and optimized.
>
> | Distribution | Solution Type | $n$ | $T$=1 | $T$=20 | $T$=40 | $T$=60 | $T$=80 | $T$=100 |
> | ---- | ---- | ---- | ---- | ---- | ---- | ---- | ---- | ---- |
> | Gaussian | Optimized | 20 | 0.2401 | 0.2830 | 0.2865 | 0.2870 | 0.2885 | 0.2892 |
> | Gaussian | Optimized | 100 | 0.1189 | 0.1401 | 0.1386 | 0.1412 | 0.1421 | 0.1427 |
> | Gaussian | Optimized | 500 | 0.0554 | 0.0619 | 0.0621 | 0.0617 | 0.0618 | 0.0612 |
> | Gaussian | Exact | 20 | 0.2183 | 0.3081 | 0.3056 | 0.3015 | 0.3014 | 0.3008 |
> | Gaussian | Exact | 100 | 0.1006 | 0.1124 | 0.1132 | 0.1136 | 0.1142 | 0.1146 |
> | Gaussian | Exact | 500 | 0.0467 | 0.0627 | 0.0643 | 0.0641 | 0.0642 | 0.0643 |
> | Exponential | Optimized | 20 | 0.1767 | 0.2284 | 0.2327 | 0.2344 | 0.2330 | 0.2323 |
> | Exponential | Optimized | 100 | 0.1009 | 0.1185 | 0.1148 | 0.1158 | 0.1173 | 0.1189 |
> | Exponential | Optimized | 500 | 0.0409 | 0.0394 | 0.0403 | 0.0402 | 0.0393 | 0.0394 |
> | Exponential | Exact | 20 | 0.1875 | 0.2445 | 0.2379 | 0.2373 | 0.2410 | 0.2405 |
> | Exponential | Exact | 100 | 0.0693 | 0.0971 | 0.0947 | 0.0951 | 0.0952 | 0.0950 |
> | Exponential | Exact | 500 | 0.0356 | 0.0432 | 0.0444 | 0.0452 | 0.0452 | 0.0456 |
> | Beta | Optimized | 20 | 0.1876 | 0.1987 | 0.2003 | 0.1987 | 0.1995 | 0.1994 |
> | Beta | Optimized | 100 | 0.0776 | 0.0967 | 0.0951 | 0.0952 | 0.0948 | 0.0943 |
> | Beta | Optimized | 500 | 0.0373 | 0.0459 | 0.0457 | 0.0453 | 0.0454 | 0.0449 |
> | Beta | Exact | 20 | 0.1767 | 0.2314 | 0.2299 | 0.2242 | 0.2216 | 0.2204 |
> | Beta | Exact | 100 | 0.0978 | 0.1096 | 0.1109 | 0.1105 | 0.1122 | 0.1121 |
> | Beta | Exact | 500 | 0.0364 | 0.0455 | 0.0457 | 0.0456 | 0.0453 | 0.0454 |
>
>
> Raw Data # 2: Comparison of model collapse for the Beta distribution described above, with various values of $\theta_0$.
>
> | $\theta_0$ | $T$=1 | $T$=20 | $T$=40 | $T$=60 | $T$=80 | $T$=100 |
> | ---- | ---- | ---- | ---- | ---- | ---- | ---- |
> | 0.1 | 1 | 2.5542 | 2.8819 | 3.0635 | 3.1824 | 3.2736 |
> | 0.2 | 1 | 2.0069 | 2.3001 | 2.4904 | 2.5923 | 2.6813 |
> | 0.5 | 1 | 1.4543 | 1.4902 | 1.4864 | 1.4935 | 1.4814 |
> | 1 | 1 | 1.1608 | 1.2274 | 1.2113 | 1.2366 | 1.2494 |

---

> > ### Comment · Reviewer_a93t · 2025-08-08
> > **Response to Authors**
> >
> > Thank the authors for the effort and time in this discussion. I appreciate the additional clarifications and supporting materials provided during the rebuttal. These have well addressed my concern, and I think this work deserves to be accepted. I will raise my score as promised.

---

### Official Review · Reviewer_aPhR · 2025-06-28

**Clarity:** 3
**Significance:** 3
**Originality:** 3
**Rating:** 5
**Confidence:** 3

**Summary:**

Consider the following method for training a generative model $p_\theta$:

a) Start with $n$ samples from the target distribution

b) Repeat:

-- Find the MLE model for all the data so far

-- Generate $n$ more samples from this learned model

Question: Does this procedure diverge from the target model?

The authors have two sets of results:

1. Positive results.

Suppose the family of generative models $p_\theta$ is very smooth, in the sense that for any $\theta$ near the target model, (i) the log density $\log p_\theta(x)$ has bounded first, second, and third derivatives, and (ii) the random variable $\log p_\theta(X)$ [for $X$ from the target distribution] is sub-Gaussian. Suppose also that the Fisher information is lower-bounded (in all directions) near the target model. Then:

-- All models ever generated by the procedure above are O(1/sqrt{n}) close to the target model in total variation distance.

2. Negative results.

The authors present a family of models that are hardly pathological (they are mixtures of uniform distributions over intervals of the real line) but do not satisfy the smoothness assumptions above. They show that although the first model generated by the learning process above is O(1/n)-close to the target distribution in total variation distance, *all* subsequent models are Omega(1)-far.

In this negative example, the target model is chosen based on the incremental sample size n. The authors also present another construction in which the target does not depend on n, and in which the models produced by the learning algorithm are Omega(1)-far from the target after f(n) iterations, where f(n) can be chosen to be as small as log log n.

**Questions:**

N/A

**Ethical Concerns:**

["NO or VERY MINOR ethics concerns only"]

**Final Justification:**

My assessment is unchanged after reading the other reviews and author feedback.

**Quality:**

4

**Strengths And Weaknesses:**

This is a timely paper, given that the data used to train generative models increasingly includes data created by prior such models. There has been a line of previous work on the same topic, and indeed the learning model used here was introduced in earlier work by Gerstgrasser et al ("Is model collapse inevitable?"). The results here complement earlier work primarily by looking at a somewhat broader class of models as well as obtaining finite-sample rather than asymptotic bounds. In particular, the negative results in this setting are a striking contrast with earlier positive results for simpler families of models (linear models and exponential families).

Overall, this is a worthwhile contribution to the literature on generative modeling.

---

> ### Author Rebuttal · Authors · 2025-07-29
>
> We thank the reviewer for their very positive feedback and their time!

---

### Official Review · Reviewer_8uXS · 2025-07-03

**Clarity:** 2
**Significance:** 3
**Originality:** 4
**Rating:** 5
**Confidence:** 3

**Summary:**

This paper studies iterative MLE, a stylised setup in which parameters are updated first with MLE on the original dataset and subsequently with MLE on the original data plus synthetically generated data from the previous iteration. The main contributions are 1) sufficient conditions under which iterative MLE estimator converges to the true parameter in l2 norm, and 2) necessary conditions, failing which the estimates with synthetic data are bounded away from the true parameter, in terms of the TV distance between the probability distributions they parameterise. In particular, the negative examples are constructed such that MLE is consistent but iterative MLE is not.

**Questions:**

Questions:
- Definition 3.1: "Given a data set $X \subset \mathcal{X}$" -- This notation would suggest that $X$ is a set of distinct elements. Perhaps the authors mean "Let $X$ be an $\mathcal{X}$-valued dataset"?
- Proof sketch of Thm 4.1: The role of Proposition C.1 is not clear in the current presentation of the proof sketch. My understanding of Proposition C.1 is that it plays role in ensuring $H_t$ is more or less stable / can be viewed as something with negligilble influence from $\theta^t$  --- can the authors confirm if this is true?
- Proof sketch of Thm 4.1: $|| \sum_{t=0}^{T-1} \frac{1}{t+1} \nabla l_t (\theta^{(t)}) \| <= O( ... )$    I think the authors want to use $ = O( ...)$ and the authors probably missed "with high probability" in this statement?
- Proof Sketch in 5.1: While I think the specific proof sketches can be shortened / partially moved to the appendix, having the explicit formula  of the distribution here is nice. However, I think the authors may want to consider having pictorial illustration of the distribution which makes it easier to see how the second iteration becomes problematic.
- Proof Sketch in 5.2: My understanding is that most of the intuitions are similar to that of 5.1 and I suggest the authors considering substantially shortening this / moving some details to the appendix, in favour of a discussion of the broader consequences of these negative examples on more practically relevant setups.

**Ethical Concerns:**

["NO or VERY MINOR ethics concerns only"]

**Final Justification:**

I was previously concerned of the significance / contribution of the paper, because I felt that the positive results were on a limited / standard theoretical model and rather expected, whereas the construction for the negative result seems very restrictive and uninformative. After the authors' clarification, I see that the negative results can be used as an example to understand how smoothness has different impacts on MLE consistency in a traditional setting vs consistency in an iterative synthetic data training setting, and I think this insight is valuable to the community. I also appreciate that the authors will add more explanations on the intuitions behind the negative examples, which will improve the paper's accessibility. Therefore I decide to raise my score to Accept.

**Limitations:**

Partially. The authors acknowlege briefly in the conclusion that the assumptions nin this work may not be the mildest possible and that more works need to be done to bridge the negative and positive results. However, I do think that even within the context of this work, to make this paper useful and interesting to the NeurIPS audience, it is important to bring out more interpretations and understanding of the currently presented negative results, which are limited to theroetical toy constructions. I am however very open to raise my scores if the authors can provide discussions or examples of this nature during the rebuttal.

**Quality:**

2

**Strengths And Weaknesses:**

Strengths: The paper is well-motivated. I find the necessary conditions (or rather, negative examples in Section 5) very interesting, and a useful complement to the understanding on model collapse. The observation that model collapse can happen even within a setup under MLE consistency is new to the best of my knowledge.


Weaknesses:
- There are some small expositions that are unclear to me but do not affect the overall flow of the paper. I have included them in the Questions section. In particular, I find that the proof sketches are not very well-presented (see Questions) and I wonder whether the authors could consider moving them to the appendix, both to gain more space in sketching the proofs in the appendix, and to have some space for elaborating on the consequences of the results.

- The sufficient conditions are expected, though interesting nevertheless. The negative examples are rather specific constructions. While the negative examples are very interesting, it would be more useful if there are some general messages that can be extracted from them -- e.g. do they inform us about verifiable conditions that may be necessary  (not necessarily in a rigorous way) to prevent model collapse? Or do we expect the irregularity of these specific constructions to manifest in more concrete, real-life ML examples? These will strengthen the impact of the paper substantially, and would have much higher value than the elaboration of the specific proofs that go behind the negative examples.

---

> ### Author Rebuttal · Authors · 2025-07-29
>
> We thank the reviewer for their constructive comments and their time. Below, we address all of the comments raised by the reviewer. If there are any concerns remaining after this response that are preventing the reviewer from raising their score, please let us know so that we can address them.
>
> - Regarding proof sketches: Thank you for pointing this out. We are happy to clarify them and add more discussion, elaborating on the consequences of the results.
>   - Role of Proposition C.1: To prove Theorem 4.1, it is important to show that when summing the differences in parameters at each time $t$, meaning $\sum_{t=1}^T \theta^{(t)} - \theta^{(t-1)}$, we need the vectors in the sum to partially cancel out so as not to compound too quickly. We follow a two-step argument,  partially inspired by classical results in the statistics literature, where we first show that each $\theta^{(t)}$ should be close enough to $\theta^{(t-1)}$ to enable a Taylor expansion, and then analyze the expansions. Proposition C.1 provides the first part, ensuring that given enough samples, the parameters of the generation $t$ model $\theta^{(t)}$ should well approximate those of the previous generation. Your intuition was correct, in that by doing this, $H_t$ can be well approximated by second-order terms, whose expected value can be expressed using the Fisher information matrix. This approximation allows us to, in turn, obtain good convergence rates for the sum.
>   - Probability in the Proof Sketch of Theorem 4.1: Indeed, the equation after line 240 is with high probability (where the explicit probability is written in the full proof). We will clarify this.
>   - Proof Sketch 5.1 and 5.2: Thanks for the suggestions! We agree that it makes sense to shorten the proof sketch of Theorem 5.2 in favor of a discussion of broader consequences. We will indeed make sure this discussion is present in the camera-ready version. We will also add pictorial illustrations to the paper to help clarify the constructions.
>
> - Interpretations and understanding the negative results: We are happy to further elaborate on this in the paper, and thank you for the suggestion. Indeed, Section 5 (the negative results) informs us about what conditions are needed to prevent model collapse. Even though the constructions are specific, they highlight more general phenomena that are needed for model collapse to occur or to be avoided. In particular, in our view, the main difference from Theorem 4.1 is the smoothness assumption. Theorem 5 crucially uses a highly non-smooth construction, in which slight perturbations of the parameters can induce huge differences in the resulting model, and we find it unlikely that a negative example would be possible without this behavior. Regarding real-life examples, even though neural networks can express very poorly behaved distributions, in many cases, they can indeed be shown to follow the smoothness assumptions, especially when using techniques like weight decay.
>
> - Definition 3.1: Thank you, this will be clarified.
>
>
>
> We hope our response addresses the reviewer’s comments, and will be happy to further clarify any point if needed.

---

> > ### Comment · Area_Chair_XUBw · 2025-08-05
> >
> > Dear reviewer,
> >
> > The discussion phase is soon coming to and end. It will be great if you could go over the rebuttal and discuss with the authors if you still have outstanding concerns. Thank you for being part of the review process.
> >
> > Regards,
> >
> > Area Chair

---

> > ### Comment · Reviewer_8uXS · 2025-08-05
> >
> > I thank the authors for the clarification, in particular the one regarding the interpretation of the negative results, which has changed my perspectives on the work. I have also gone through the author's discussion with other reviewers. I think messages along the lines of "model collapse can happen despite MLE consistency due to non-smoothness" / "iterative synthetic data training can be a lot more sensitive to non-smoothness than MLE" are indeed very nice and it'd be great to see them highlighted in the Discussion / towards the end of Section 5. I decide to raise my score as a result.

---

> > > ### Author Response · Authors · 2025-08-05
> > > **Reviewer 8uXS**
> > >
> > > We thank the reviewer for their encouraging response, and will indeed highlight these points in the paper.

---

### Decision · Program_Chairs · 2025-09-17

**Decision:**

Accept (poster)

**Comment:**

This paper offers a solid theoretical analysis of iterative MLE in the context of model collapse. It establishes sufficient conditions under which iterative MLE remains consistent, while also presenting carefully constructed counterexamples that demonstrate how inconsistency can arise in non-smooth settings. These negative results are not only technically interesting but also provide valuable insights into why smoothness plays a crucial role in preventing collapse, thereby enriching our understanding of generative model training on synthetic data. The work is clearly written and well motivated, and reviewers uniformly provided positive scores. I recommend acceptance as a poster.